# Romanian Bee Product Analysis: Chemical Composition, Antimicrobial Activity, and Molecular Docking Insights

**DOI:** 10.3390/foods13101455

**Published:** 2024-05-08

**Authors:** Silvia Pătruică, Suleiman Mukhtar Adeiza, Anca Hulea, Ersilia Alexa, Ileana Cocan, Dragos Moraru, Ilinca Imbrea, Doris Floares, Ioan Pet, Florin Imbrea, Diana Obiștioiu

**Affiliations:** 1Faculty of Bioengineering of Animal Resources, University of Life Sciences “King Mihai I” from Timișoara, Calea Aradului nr. 119, 300645 Timisoara, Romania; silviapatruica@usvt.ro (S.P.); dragos.moraru@usvt.ro (D.M.); ioanpet@usvt.ro (I.P.); 2Faculty of Life Science, Department of Biochemistry, Ahmadu Bello University, Zaria 810107, Kaduna State, Nigeria; 3Faculty of Veterinary Medicine, University of Life Sciences “King Mihai I” from Timișoara, Calea Aradului no. 119, 300645 Timisoara, Romania; doris.oarga@usvt.ro (D.F.); florin_imbrea@usvt.ro (F.I.); dianaobistioiu@usvt.ro (D.O.); 4Faculty of Food Engineering, University of Life Sciences “King Mihai I” from Timișoara, Calea Aradului no. 119, 300645 Timisoara, Romaniaileanacocan@usvt.ro (I.C.); 5Faculty of Agriculture, University of Life Sciences “King Mihai I” from Timișoara, Calea Aradului no. 119, 300645 Timisoara, Romania; ilinca_imbrea@usvt.ro

**Keywords:** propolis, royal jelly, apilarnin, LC-MS, microdilution, molecular docking

## Abstract

Bee products are considered true wonders of nature, used since ancient times, and studied even today for their various biological activities. In this study, we hypothesise that Romanian bee products from different origins (micro apiary products, lyophilised forms, commercial) exhibit distinct chemical compositions, influencing their biological activities. An LC-MS analysis revealed varied polyphenolic content patterns, with cumaric acid, ferulic acid, rosmarinic acid, and quercitine identified in significant amounts across all samples. Primary anti-inflammatory evaluation phases, including the inhibition of haemolysis values and protein denaturation, unveiled a range of protective effects on red blood cells (RBC) and blood proteins, contingent upon the sample concentration. Antimicrobial activity assessments against 12 ATCC strains and 6 pathogenic isolates demonstrated varying efficacy, with propolis samples showing low efficacy, royal jelly forms displaying moderate effectiveness, and apilarnin forms exhibiting good inhibitory activity, mostly against Gram-positive bacteria. Notably, the lyophilised form emerged as the most promising sample, yielding the best results across the biological activities assessed. Furthermore, molecular docking was employed to elucidate the inhibitory potential of compounds identified from these bee products by targeting putative bacterial and fungal proteins. Results from the docking analysis showed rosmarinic and rutin exhibited strong binding energies and interactions with the putative antimicrobial proteins of bacteria (−9.7 kcal/mol to −7.6 kcal/mol) and fungi (−9.5 kcal/mol to −8.1 kcal/mol). The findings in this study support the use of bee products for antimicrobial purposes in a biologically active and eco-friendly proportion while providing valuable insights into their mechanism of action.

## 1. Introduction

The primary concerns of researchers in recent years are directed towards finding alternative methods of prevention and treatment of microbial infections [1], justified by the appearance of multidrug-resistant bacterial strains [2]. The concern is more significant since the pharmaceutical industry has not made a new antibiotic available in the last ten years, and the abusive and unjustified use of antimicrobials has led to the emergence of bacterial strains resistant to the majority of antibiotics used in medical practice. Recent studies show that natural products not only have an antimicrobial effect against numerous bacterial species [3,4,5,6], but also do not cause the appearance of resistance, so they would be the ideal candidate as an alternative treatment method to the detriment of antibiotics. Of all natural products, bee products are considered true wonders of nature, used since ancient times, and studied even today for their various biological activities, such as antibacterial, anti-angiogenic, antiulcer, anti-inflammatory, antioxidant, and antiviral activities [4,7,8,9].

Propolis, also called bee glue, represents a sticky, resinous substance produced by honeybees to defend the hive, but it has also found its use in natural medicine since ancient times. Due to its composition, a mixture of natural substances, such as plant resin, wax, essential and aromatic oil, pollen, and other natural products [7,10], propolis has a wide range of health benefits [7]. However, raw propolis cannot be used directly; it requires an extraction process in various solvents, through which approximately 70% of the components are released [11]. As such, its bioactivity is variable. This variability is due not only to the extraction process or types of solvent used [12,13], but also to the chemical composition of raw propolis, which varies depending on the geographical area (altitude) and, respectively, the climate, illumination, and seasonal variations [7,10,11]. Of all the chemical compounds of propolis, flavonoids and phenolic compounds play an essential role in antibacterial activity [7,10,11]. The antimicrobial activity was demonstrated against both Gram-positive (*S. aureus*, *S. epidermidis*, *S. mutans*) and Gram-negative bacteria (*E. coli*, *Pseudomonas aeruginosa*, *Klebsiella pneumoniae*), of course, with variations in the minimum inhibitory concentration, and variations justified by the different chemical compositions [14,15,16]. However, the antibiofilm activity represents propolis’s most crucial clinical benefit [17].

Royal jelly (RJ), a glandular secretion produced from young nurses’ hypopharyngeal and mandibular salivary glands, represents the exclusive nourishment for bee larvae which are selected to become queens [18]. Over time, RJ has demonstrated many biological activities in the medical field, such as antitumour, anti-allergy, anti-inflammatory, and antimicrobial activities [19]. The antimicrobial activity seems to be directed towards both Gram-negative and Gram-positive bacteria, respectively, against *S. intermedius*, *L. monocytogenes*, *S. epidermidis*, *E. coli*, *Enterobacter cloacae*, *Klebsiella pneumoniae*, and *P. aeruginosa* [20,21,22]. Moreover, studies demonstrate an increased effectiveness of RJ against aerobic and anaerobic bacteria involved in periodontal diseases [23]. Like other bee products, this activity is due to the phenolic compounds and flavonoids [24].

A drone brood, also called apilarnil, is a bee product obtained from the collection of drone larvae [25]. In the literature, there are little data about this product. Most of the data refer to its complex chemical composition, consisting of protein (9–12%), including amino acids, carbohydrates (6–10%), lipids (5–8%), vitamins, and minerals [9,26]. Among the biological activities, it stands out for its antioxidant, antiatherosclerotic, androgenic, and adaptogenic effects [25]; as far as we know, antimicrobial activity is not being studied.

Romania is a significant exporter of honey and other bee products, being ranked among the first because the temperate continental climate and the variety of melliferous plants in the Carpathian-Danubian-Pontic area offer favourable conditions for apiculture [27]. The high production is ensured by around 40,000 beekeepers with 1,550,000 bee colonies, spread across all 41 counties of the country, with a density of approximately 6.2 colonies/km^2^. The predominant species comprises *Apis mellifera carpathica*, and some other species are imported [28]. Regarding resin sources of Romanian propolis, the primary botanical origin consists of Populus, Ulmus, Quercus, Salix, Aesculus, and Fraxinus species. The most common compounds found are caffeic acid, CAPE (caffeic acid phenethyl ester), chrysin, and pinocembrin, but also ferulic and p-coumaric acids, kaempferol, galanin, and quercetin [29,30,31]. The chemical composition of Romanian royal jelly and apilarnil is characterised by high concentrations of rutin, chlorogenic acid, and isoquercitrin. In addition, apilarnil contains an increased amount of naringenine, while royal jelly does not. In contrast to propolis, cafe-ic and p-coumaric acids are detected in low concentrations [32].

Furthermore, in silico molecular modelling has been shown to give insight into the antimicrobial activities of docked phytocompounds in the crystal structure (macro-molecular targets) of bacteria and fungi. These biologically active compounds effectively allow the mechanism of cell wall disruption of the microbial pathogens to be biochemically elucidated [33]. Bacterial proteins such as tyrosyl-tRNA synthetase (PDB: 1JII) are identified to be an interesting target for drug-resistant bacteria [34], and peptide deformylase (PDB: 1G27) has gained attention as a novel attractive target of major antibiotics due to its mode of catalysis involving the removal of N-formylmethionine as the initiator of newly synthesised polypeptides, which is a crucial step in the maturation of bacterial protein [35]. DNA gyrase (PDB: 3LPX), a type II topoisomerase, plays a vital role in introducing negative supercoils into DNA in the presence of ATP [36], thereby changing the topology of the DNA. This supercoiling mechanism is the essential nature of DNA gyrase, making the enzyme a potential drug target for antibacterial and anticancer chemotherapy [37]. On the other hand, fungal targets involving squalene epoxidase (PDB: 2QA1) are an attractive potential target for inhibiting the growth of pathogenic fungi [38].

In contrast, fungal cell walls are mainly composed of β-glucans, which is a homopolysaccharide of β-1,3-linkage along with varying proportions of β-1,6- and β-1,4-linked glucose side chains [39]. The enzyme 1,3-β-glucan synthase (PDB: 8JZN) is a multi-subunit complex that contributes to the synthesis of the fungal wall assembly. N-myristoyl transferase (PDB: 5AG7) is abundant in the membrane/wall of eucaryotic organisms (protozoa, fungi, mammals). It catalyses the co-translational and/or post-translational reactions involving the attachment of myristate from myristoyl coenzyme A to the N-terminal glycine residue of a fungal protein via a covalent bond [40,41].

The main objectives of this study were to evaluate and compare the chemical composition of some commercial bee products and those collected directly from a micro apiary (royal jelly, apilarnil, and propolis) located in the western part of Romania, in a natural state or lyophilised. Moreover, the antimicrobial activity of the studied bee products was studied against some standardised Gram-negative and Gram-positive bacterial strains and fungal strains, as well as some pathogenic isolated strains isolated and identified previously from pyogenic dermatitis lesions in dogs. Molecular docking was performed in this study to show the inhibitory potential of the compounds identified from these bee products by targeting putative bacterial and fungal proteins. Docking can predict the binding affinities and possible interaction models of ligands and protein targets [42].

## 2. Materials and Methods

### 2.1. Samples of Bee Products

The study was conducted on three types of bee products: propolis, royal jelly, and apilarnil. Both commercial and fresh micro apiary samples were taken into the study. The fresh samples were collected from a micro apiary (small cluster of colonies (hives) in different locations) belonging to USVT located in Caransebeș County, Banat region, Romania (45024′48.6″ N 22012′53.7″ E). All the samples were stored in glass containers at 0 ± 5 °C before being analysed. As mentioned above, in terms of propolis, the analysed samples were collected from the USVT micro apiary or commercially available samples. To prepare the propolis tincture, 20 g of raw propolis was macerated in 100 mL ethanol (70%).

Regarding royal jelly, freeze-dried and pure forms were tested with commercial and freshly harvested origins. Similarly, freeze-dried and fresh apilarnil samples were subjected to the study. It should be noted that the lyophilised apilarnil was obtained from the commercial pure one. Lyophilisation was carried out by using the Unicryo MC4L −60 °C lyophiliser (Uniequip, Martinsired, Germany). Concerning the microbiological analysis, aqueous extracts from each sample were prepared at a 1:10 ratio using sterile distilled water, shaken for 30 min, and filtered. Subsequently, different quantities were spotted in 96-well plates. The abbreviations used are as follows: propolis from micro apiary source, MASP; commercial propolis, CP; propolis tincture, PT; royal jelly from micro apiary source, MASRJ; commercial lyophilised royal jelly, CLRJ; commercial royal jelly, CRJ; apilarnil from micro apiary source, MASA; commercial lyophilised apilarnil, CLA; commercial apilarnil, CA.

### 2.2. Individual Polyphenols Content Detected by Liquid Chromatography-Mass Spectrometry (LC-MS)

The LC-MS method for determining individual polyphenol content was used according to the procedure described by Cadariu et al. [43]. The equipment was LC-MS (Shimadzu 2010 EV, Kyoto, Japan) equipped with electrospray ionisation and SPD-10A UV and LC-MS 2010 detectors. Chromatography conditions for the determination of polyphenolic compounds were as follows: Nucleodur CE 150/2 C18 Gravity SB column 150 mm × 2.0 mm (Macherey-Nagel GmbH & Co. KG, Düren, Germany) and a flow rate of 0.2 mL/min. The compounds under determination were separated with an elution gradient: 5% B (0.01–20 min), 5–40% B (20.01–50 min 10 min), 40–95% B (50–55 min), and 95% B (55–60 min), where B is acetonitrile and formic acid solution, pH = 3. Calibration curves used were in the range of 20–50 g/mL. Individual polyphenolic compounds detected in the 9 samples were expressed as µg/g dry weight (d.w.). The individual polyphenols analysed were epicatechin, caffeic acid, β-rezolcilic acid, cumaric acid, rutin, ferulic acid, rosmarinic acid, resveratrol, and quercitin. All samples were analysed in triplicate, and results were presented as mean ± standard deviation (SD).

### 2.3. Determination of the Antimicrobial Activity

The antimicrobial activity of the samples was determined by broth microdilution against Gram-positive, Gram-negative, and fungal ATCC strains and pathogenic isolates, isolated and identified before the present research from pyogenic dermatitis in dogs. These drug-resistant strains are part of the culture collection of the Laboratory of Microbiology culture collection in the Interdisciplinary Research Platform within the University of Life Sciences “King Mihai I of Romania” Timisoara.

The tested strains were as follows: Streptococcus pyogenes (ATCC 19615), Staphylococcus aureus (ATCC 25923), Shigella flexneri (ATCC 12022), Pseudomonas aeruginosa (ATCC 27853), Escherichia coli (ATCC 25922), Salmonella typhimurium (ATCC 14028), Haemophilus influenzae tip B (ATCC 10211), Bacillus cereus (ATCC 10876), C. perfringens (ATCC 13124), Listeria monocytogenes (ATCC 19114), Candida albicans (ATCC 10231), and Candida parapsilopsis (ATCC 22019).

The samples were also tested against clinically drug-resistant strains isolated from pyogenic dermatitis. In our laboratories, the strains were maintained at −50 °C. After the laboratory acronym, isolates from pyogenic dermatitis are abbreviated with MLIRP, followed by the number of strains: *E. coli* (MLIRP code 062019), *E. coli* (MLIRP code 022020), *E. coli* (MLIRP code 112020), *P. aeruginosa* (MLIRP code 122021), *P. aeruginosa* (MLIRP code 092022), *P. aeruginosa* (MLIRP code 042019), *S. aureus* (MLIRP code 092020), *S. aureus* (MLIRP code 072022), and *S. aureus* (MLIRP code 052022).

#### 2.3.1. Bacterial Culture

Our previous study describes the methods used by Dégi et al. [44] and Obistioiu et al. [45]. The concentrations tested were selected based on prior research and a literature search to cover a broad range of concentrations and find possible MIC values [4,8,19,45].

The extracts were used directly by adding 0.25, 0.5, 0.75, 1.0, 2.5, 5.0, 7.5, or 10.0 mg/mL to the bacterial suspension. A pure uninhibited strain in BHI was used as a positive control, and the value was subsequently used to calculate the bacterial growth and inhibition rates.

The MIC was determined by the measurement of OD using the spectrophotometric method, according to [46]. The MIC is the lowest compound concentration that yields no visible microorganism growth.

BGR (bacterial growth rate) and BIR (bacterial inhibition rate) were calculated as indicators for interpreting the results using the following formulas:
$$\mathrm{BGR}=\frac{OD_{sample}}{OD_{negative\;control}}\times 100\;(\%)$$
 BIR = 100 − BGR (%)
where OD sample is the optical density at 540 nm as a mean value of triplicate readings;

OD negative control is the optical density at 540 nm as a mean value of triplicate readings for the selected bacteria in BHI.

#### 2.3.2. Fungal Culture

The analysis was conducted according to our previous research [45], with small modifications regarding the quantity tested. The extracts were used directly by adding 0.25, 0.5, 0.75, 1, 2.5, 5, 7.5, or 10 mg/mL over bacterial suspension. The plates were incubated for 48 h at 37 °C. After incubation, the OD was measured at 540 nm. All samples were read in triplicate.

The following formulas were used to calculate MGR (mycelial growth rate) and MIR (mycelial inhibition rate):
$$\mathrm{MGR}=\frac{OD_{sample}}{OD_{negative\;control}}\times 100\;(\%)$$
MIR = 100 − MGR (%)
where OD sample is the optical density at 540 nm as a mean value of triplicate readings;

OD negative control is the optical density at 540 nm as a mean value of triplicate readings for the selected fungi in BHI.

### 2.4. Molecular Docking Study

The interactions between the identified compounds from 3.11 and the crystallographic structure of bacterial proteins (PDB: 1JII, 1G27, 3LPX) and fungal proteins (PDB: 2QA1, 8JZN, 5AG7) obtained from the Protein Data Bank (PDB) [47] were assessed via the docking analysis to predict the possible binding conformation of the ligands (compounds) and the receptor (proteins). The proteins were prepared for docking using the UCSF Chimera 1.17.3 DockPrep tool, which involves the removal of water molecules co-crystallised ligand/s while AutoDock polar hydrogens and Gasteiger charges were added to the protein structure. The receptor docking grid was defined in PyRx software (Python Prescription 0.8) to maximum coverage, which works with AutoDock vina (AutoDockTools 1.5.2). For the bacterial protein 1JII, the gid box coordinates were interface in centre X = −11.879, Y = 17.2390, Z = 91.1176, and dimension (angstrom) X = 44.7763, Y = 66.3095, Z = 51.5355. For 1G27, centre X = 44.7723, Y = 0.1033, Z = 17.6607, and dimension (angstrom) X = 44.8834, Y = 50.8513, Z = 38.7750. For 3LPX, centre X = 17.8737, Y = 94.9472, Z = 9.42.97, and dimension (angstrom) X = 57.1074, 102.1072, Z = 75.6027. While for the fungal protein 2AQ1, centre X = 28.0408, Y = 61.0097, Z = 80.4050, and dimension (angstrom) X = 56.1540, Y = 68.5502, Z = 65.4575. For 8JZN, centre X = 157.250, Y = 160.4523, Z = 147.4263, and dimension (angstrom) X = 90.7495, Y = 104.7751, Z = 121.5846. For 5AG7, centre X = 18.5697, Y = 0.7541, Z = 7.6232, and dimension (angstrom) X = 59.2264, Y = 68.7556, Z = 65.6434.

The three-dimensional and two-dimensional interactions of the receptor ligands were visualised and analysed by Discovery Studio Visualizer v21.1.0.20298 software.

## 3. Results

### 3.1. Individual Polyphenols Content Detected by LC-MS

Table 1 shows the chromatographic profile of individual polyphenols separated by LC-MS from the nine royal jelly, apilarnil, and propolis product extracts.

Four compounds were identified in the highest (but variable) amounts from all samples: cumaric acid, ferulic acid, rosmarinic acid, and quercitine.

Ferulic acid and resveratrol were the main polyphenolic compounds identified in propolis extracts. The highest value was recorded for ferulic acid in CP (198.72 ± 12.47 µg/g) and resveratrol in the MASP sample (137.70 ± 3.44 µg/g). The percentage of rosmarinic acid of all polyphenols identified in propolis had values between 67.9 and 87.50 µg/g. Quercitin was also consistently present in varying concentrations in all nine samples (Table 1), accounting for 15.60 and 16.80 µg/g. The lowest concentration was in MASP (15.60 ± 0.39 µg/g), while the highest was in CP (16.80 ± 1.92 µg/g).

For royal jelly, the following individual polyphenols were identified in all samples: epicatechin (73.80–74.20 µg/g), coumaric acid (39.10–39.60 µg/g), ferulic acid (127.80–129.60 µg/g), rosmarinic acid (68.10–69.10 µg/g), resveratrol (119.10–233.30 µg/g), and quercitin (15.7–15.9 µg/g), and caffeic acid and rutin were undetermined.

In the case of apilarnil extracts, it could be observed that β-rezolcilic acid had the highest amount in the determinations (175.10 ± 4.38 µg/g in MASA and CLA samples), followed by ferulic acid (126.36 ± 3.16 µg/g in the MASA sample), epicatechin with a maximum concentration of 74.20 ± 1.86 µg/g in the CLA sample, and gallic acid could not be determined in these extracts. These are the most relevant findings of the present study for apilarnil, as no other Romanian apilarnil types have been determined.

### 3.2. Antimicrobial Activity

Table 2 presents the antimicrobial activity as the bacterial inhibition rate (BIR%) values against Gram-positive ATCC strains. The Appendix A present the results in graphical form: the bacterial growth rate (BGR%) and bacterial inhibition rate (BIR%) for each strain (Appendix A).

Concerning *S. pyogenes* ATCC, the results show a negative picture from the inhibition point of view. The only positive results were obtained in the MASRJ, MLL, MCL, and CLA samples. All the samples showing positive results occurred only at high quantities tested above 2.5 mg/mL. The results show a low efficacy against *S. pyogenes* in the case of the four less efficient MASRJ, CLRJ, CLJ, and CLA samples. All the other samples proved a strain-boosting effect, which was demonstrated by the increase in the optical density.

Regarding the antimicrobial activity against *S. aureus*, five of the tested samples showed no effect regarding inhibition (Table 2 and Appendix A). MASP, CP, PT, MASA, and AP showed a strain-boosting impact, but the trend was negative, with a negative correlation. The least effective of the samples tested proved to be MASA, with a negative value regarding BIR% of −241.80%, followed by PT, AP, and MASP. However, the values obtained were still higher than those obtained for the control; therefore, they remained effective. MASP and AP proved to have inhibitory values that were positive only concerning the smaller quantity tested, with values that reached only the 2.5 mg/mL quantity tested. CLRJ developed antibacterial efficiency but only starting from 1 mg/mL. MASRJ, CLJ, and CLA were the samples identified as having good antibacterial efficiency. BIR% reached levels varying from 22.79% to 79.01% in their cases.

*B. cereus* presented components of the bacterial wall that formed a synergy with elements from the samples tested (Table 2 and Appendix A), with the effect being strain-boosting in most of the extracts tested. A negative correlation with the increase in quantity was found in CP, PT, MASA, and AP. MASP demonstrated MIC only at 10 mg/mL tested, CLJ at 5 mg/mL, and CLRJ at 0.5 mg/mL. MASRJ and CLA were the only samples that proved efficacy, in positive correlation with the increased quantity tested, with values of BIR% ranging from 68.80% to 78.03%.

Table 2 (and Appendix A) presents the antimicrobial activity of the samples tested against *Cl. perfringens*. The overall picture is one of low efficacy, with most of the samples proving a strain-boosting effect: MASP, CP, PT, MASA, CLA (in which, even if the trend is negative, the values are positive, reaching MIC at the first quantity tested), and AP. CLJ and CLRJ showed negative values at the first quantities tested, so MASRJ was the only one with a positive effect.

*L. monocytogenes* responded well to the activity of the tested samples (Table 2 and Appendix A). MASA and AP were the only two that were affected of the first quantities tested.

Table 3 presents the antimicrobial activity as bacterial inhibition rate (BIR%) values against the Gram-negative bacteria and two *Candida* spp. ATCC strains. The Appendix A present the results in graphical form: the bacterial growth rate (BGR%) and bacterial inhibition rate (BIR%) for each strain (Appendix A).

In *S. flexneri*, the effectiveness percentage was similar to those obtained in the case of *S. aureus*, as presented in Table 3 and Appendix A. MASP demonstrated negative inhibition values, but with a positive trend, the values increased with increasing quantity. This indicates that the amounts tested had not yet reached the value required to cause an adequate inhibitory effect. PT showed that from the beginning, the activity was to stimulate bacterial growth, and the inhibition rate decreased with the increase in quantity, reaching values of −100.69%. CP, MASRJ, CLJ, CLRJ, and CLA have demonstrated antibacterial efficacy, with the most effective being shown to be RJ, followed by CLA and CML. MASA and AP showed negative inhibition trends with efficacy only in the first four and five quantities tested.

*P. aeruginosa* proved to be one of the strains most resistant to the action of the tested samples, with antibacterial activity being identified only in the case of samples MASRJ, CLJ, CLRJ, and CLA (Appendix A). Even in the case of these samples, positive trends correlated with the increase in quantity, which was present only in CLRJ and AML. All other samples had a growth-stimulating effect on the bacterial strain, with inhibition values up to −600.55% in the MASA sample and −394.90% in the PT sample. This demonstrates 500% higher bacterial growth in the MASA sample and 294% higher in the PT sample compared to the control.

When analysing the results obtained from the point of view of the samples tested against the *E. coli* strain, the picture presented has a higher efficacy than average (Table 3 and Appendix A). Thus, the efficacy of these samples classified ascendingly was MASP, CP, CLRJ, MASRJ, and CLJ. PT, MASA, and AP showed a negative trend, stimulating the bacterial strain. CLA also demonstrated a strain-boosting effect, but the values obtained were superior to those of the control and can be classified as effective.

Table 3 (Appendix A) presents the results obtained in the antibacterial efficiency analysis of samples tested against *S. typhimurium.* The general picture is one with an efficacy above average, and a negative trend correlated with the increase in quantity found only in the case of the MASP, PT, MASA, and AP samples. In the case of the MASP sample, the values obtained, although decreasing, were positive from 50.17% and reduced to 12.96% in the case of the maximum tested quantity. All other samples tested demonstrated very good efficacy, with the maximum values recorded at the last amount tested ranging from 60% to 89.37%.

Concerning the antibacterial efficacy of the tested samples, Table 3 (Appendix A) presents BIR% values obtained regarding *H. influenzae*. MASP, PT, MASA, and AP presented a negative correlation with the increase in the quantity of the inhibition values. The other samples demonstrated good efficiency in the following decreasing order: CLA > CLJ > MASRJ > CLRJ > CP.

Regarding the antifungal effect, *C. albicans* proved resistant to all the tested samples. Still, MASRJ, CLJ, CLRJ, and CLA developed small inhibition percentages and tested only at high quantities (Appendix A and Table 3). *C. parapsilopsis* appeared more susceptible to the samples’ antifungal effect (Appendix A). MASA and AP were the only samples with a negative inhibitory trend; all the other samples tested demonstrated good efficacy, with BIR% values reaching maximum values ranging from 16.16% to 90.51%.

Table 4 presents the inhibitory (%) values obtained for the analysed samples against the clinical isolates selected.

Regarding the bee products’ activity against wild *Staphylococcus aureus* isolates (Appendix A), the data are presented in Table 4. MLIRP 052022 proved to have different responses but with low sensitivity to the products tested. Propolis demonstrated a strain-boosting effect; the inhibitory values increased alongside the increase in concentration. MASRJ, CLRJ, and CLJ proved to have an inhibitory pattern, an evolution with a positive trend, but MIC reached only at 7.5 mg/mL (MASRJ), 0.5 mg/mL (CLRJ), and 10 mg/mL in the case of CLJ (Table 2). A similar response was achieved in MLIRP 052022 after the action of apilarnin in all three forms, with MIC ranging from 2.5 mg/mL to 7.5 mg/mL. A better inhibitory activity was present in the case of *S. aureus* MLIRP 072022, against which propolis had only strain-boosting activity. Still, the other values obtained presented the following efficacy: CLA > MASRJ > MASA > CLRJ > CLJ > AP. Concerning *S. aureus* MLIRP 092020, the resistance pattern was different. MASP, CP, and PT showed inhibitory evolutions, but MIC was present only at 10 mg/mL (MASP) or MIC was not reached within the concentration ranges tested (CP and PT). On the contrary, CLRJ demonstrated a strain-boosting effect, with BIR% values ranging from −0.20% to −18.27%, with an unfavourable evolution, and the inhibition values (%) decreasing with the increase in concentration. Apilarnin was the only bee product with a good inhibitory response, with the MIC varying from 0.25 to 1 mg/mL.

Concerning *P*. *aeruginosa* pathogenic isolates, all three isolates proved to have a high resistance, and all the bee products at all the concentrations tested demonstrated a strain-boosting effect (Appendix A). Even if the first concentrations tested inhibited in a small percentage the strain growth, with BIR% values ranging from 9.10 to 0.05% at 0.25 mg/mL (Table 4), the last concentration tested reached BIR% values varying from −18.95% to −2.10%. The only exception was CLRJ, which, in the case of *P. aeruginosa* MLIRP 122021, registered, even if with a negative evolution sustaining a strain-boosting activity, only positive values starting from 10.10% at 0.25 mg/mL to 0.67% at 10 mg/mL.

*E. coli* pathogenic isolates demonstrated resistance (Appendix A) to MASP, CP, and PT, with only *E. coli* MLIRP 062019 reacting with positive values, although with a negative strain-boosting trend. CLJ proved medium efficacy BIR% values, albeit in decreasing order, as did MASRJ and CLRJ, but in 66% of the strains tested. At the same time, apilarnin demonstrated good efficacy, with a favourable evolution trend in all three forms tested (own production, lyophilised, and commercial), with MIC ranging from 0.25 mg/mL to 2.5 mg/mL.

Table 5 presents the MIC values obtained against the ATCC strains and pathogenic isolates.

### 3.3. Inhibition of Haemolysis Values and Protein Denaturation

Inhibition of Haemolysis Values

The results obtained by the heat-induced haemolysis assay are presented in Table 6. It was observed that MASP, CP, and CLRJ had a protective effect against haemolysis starting from 0.25 mg/mL; the percentage of haemolysis inhibition increased values with the concentration of samples. Of the three, at a concentration of 25%, CP had the highest value of haemolysis inhibition percentage, respectively, of 47.22%, reaching 58.83% at the highest concentration tested. Similarly, CLRJ showed a high value of the haemolysis inhibition percentage at a concentration of 25%, respectively, of 41.56%, reaching a value slightly over 50% at the maximum tested concentration. The percentage of inhibition of the haemolysis value for CP at a concentration of 25% was only 0.75%, reaching 18.71% at a concentration of 10%. The samples demonstrated the anti-inflammatory activity by the membrane lysis assay in the following order: MASP > CLRJ > CP. However, all these samples showed that a concentration of 10% determined the inhibition percentage values close to those obtained by using 0.1 mg/mL dexamethasone. Regarding the other samples, only CLJ and AP demonstrated the protective effect against the membrane lysis at the highest tested concentration for the rest of the samples.

Inhibition of Protein Denaturation

Table 6 presents the results for the effect on protein denaturation. From all the studied samples, MASP and CP demonstrated that the percentage inhibition of protein denaturation values followed an ascending curve with minimum values at a concentration of 2.5% and maximum values at 10%.

For the other samples, the percentage inhibition of protein denaturation followed a descending slope, with maximum values at the minimum tested concentration of the samples. However, even at the maximum concentration tested (10%), the inhibition of protein denaturation values was more than 50% for CLRJ, CLJ, and AP.

### 3.4. Docking Analysis

Molecular docking has contributed immensely to the downstream techniques of rational drug design and has become pharmaceutically relevant [48,49]. This study showed, in a virtual model, the possible intramolecular binding modes between the identified polyphenols and the proteins of microbial pathogens. The ten (10) compounds mainly detected were docked into the respective crystal structures of the bacterial and fungal proteins (Table 7 and Table 8).

The analysis further revealed how the compounds were held in the binding pocket through defined hydrogen bonds and hydrophobic and steric interaction forces, and most importantly, the anchorage of hydrogen bonds exhibited a potent antimicrobial activity through their binding modes (Figure 1, Figure 2, Figure 3, Figure 4, Figure 5 and Figure 6). The ternary structure of bacterial tyrosyl-tRNA synthetase had binding residues of CYS37, GLY38, ALA39, ASP40, HIS47, GLY49, HIS50, LEU70, THR75, GLN174, ASP177, GLN190, GLY192, ASP195, and PRO222 [42]. This conforms with our docking results, which were predicted by caffeic acid (−7.0 kcal/mol), gallic acid (−7.2 kcal/mol), resveratrol (−8.2 kcal/mol), and rutin (−9.7 kcal/mol) binding to these amino acid residues (Figure 1). These binding energies are indicators of a strong interaction between the compounds, and critical amino acids of the protein-binding pocket stabilize tyrosyl-tRNA synthetase. In the same fashion, epicatechin (−7.8 kcal/mol), ferulic acid (−6.0 kcal/mol), gallic acid (−5.9 kcal/mol), and rosmarinic acid (−8.2 kcal/mol) interacted nicely with the amino acid residues GLN50, GLU133, GLY81, LEU91, GLN65, LEU112, VAL50, and GLY110 in the binding pocket of the protein (Figure 2). This was visibly stable in the complex formation between these amino acids and the compounds of the bee products, owing to more hydrogen bonds being formed and fewer hydrophobic interactions. Seemingly, the binding affinity of the compounds was in fit conformation to the amino acids of peptide deformylase. The scoring energy for caffeic acid (−6.2 kcal/mol), quercetin (−7.9 kcal/mol), rutin (−9.0 kcal/mol), and rosmarinic acid (−7.6 kcal/mol) with high affinity to DNA gyrase indicated the possible number of DNA unwinding sites for their interaction with the protein, which was evident from the number of hydrogen bonds complexed with the crystal structure of DNA gyrase (Table 3). These deductions further strengthened the biological activities of these compounds to potentially halt the activity of these targeted enzymes in bacterial protein biosynthesis and DNA unwinding potential.

Among the fine interactions of the compounds and fungal proteins, gallic acid and rosmarinic acid formed more hydrogen bonds with three therapeutic targets (2QA1, 8JZN, 5AG7) in a sum that interacted with the critical amino acid residues in the active sites of these proteins (Figure 3, Figure 4, Figure 5 and Figure 6). Squalene epoxidase maintained a firm interaction with gallic acid (−6.8 kcal/mol), forming the following: hydrogen bonds with LEU31, GLU32, ARG33, and VAL120; carbon-hydrogen bonds with GLY9; and a pi-alkyl bond with VAL8. At best, the possible interaction for 1,3-β-glucan synthase was seen with rutin (−9.4 kcal/mol). Five hydrogen bonds, two carbon-hydrogen bonds, and four pi hydrophobic bonds constituted a favourable binding affinity in the receptor–ligand complex. On the other hand, quercetin (−8.7 kcal/mol) and resveratrol (−7.8 kcal/mol) formed one hydrogen bond when docked with the protein, among other hydrophobic bonds. This reflects the weak and unfavourable interaction even though they showed high binding affinity to the protein. The binding modes of gallic acid and N-myristoyl transferase imply a suitable complex formation with the protein crystal structure (Figure 6), showing the binding energy at −6.5 kcal/mol occupied five hydrogen bonds in the protein binding site while presenting other hydrophobic interactions (VAL419 and VAL378).

Another hydrophobic interaction was observed with the other compounds having only one hydrogen bond with the protein. Consequently, the weak intramolecular interactions reflect the unfavourable binding energies observed in both the docking outcomes of the bacterial and fungal proteins, particularly for the compound p-coumaric, which had seemingly low binding energies and hydrogen bond interactions across the proteins.

Interesting matches were observed between the binding energies and interactions with caffeic acid and tyrosyl-tRNA synthetase (−7.0 kcal/mol); epicatechin and peptide deformylase (−7.8 kcal/mol); DNA gyrase (−7.5 kcal/mol), gallic acid, and tyrosyl-tRNA synthetase (−7.2 kcal/mol); quercetin and DNA gyrase (−7.9 kcal/mol); and resveratrol and tyrosyl-tRNA synthetase (−8.2 kcal/mol). Rosmarinic acid and rutin had high binding energies (−7.6 kcal/mol −9.7 kcal/mol) and high interactions across the putative bacterial proteins. Additionally, an almost similar result was visualised between the rosmarinic acid, rutin, and fungal proteins, −8.1 kcal/mol–−9.5 kcal/mol, except in the interaction of rutin and squalene epoxidase (−9.4 kcal/mol), which had a single hydrogen bond and other hydrophobic interactions.

## 4. Discussion

### 4.1. Individual Polyphenols Content Detected by LC-MS

Studies demonstrate that the biological activity of bee products is dependent on their chemical composition, on the total content of phenols and flavonoids, respectively, and on the phenolic profile, which varies depending on bee species, the geographical origin, the harvesting season, and the methods of extraction [50,51,52,53,54].

Potential floral sources of propolis vary depending on the geographical area [55,56,57], which determines the variable chemical composition of this bee product. For example, most propolis samples collected from various regions of Poland were chemically characterised by the predominance of epicatechin/catechin, pinobanksin, and myricetin, with significant concentration variations from one sample to another. The phenolic acid concentrations determined were low, with the predominance of syringic acid, followed by ferulic acid [50]. Instead, by analysis, the Moroccan and Palestinian propolis were observed with a high variation among the samples: four of the five studied samples had high concentrations of flavonoids, respectively, pinocembrin, naringenin, and chrysin, and one of them was characterised by a high concentration of cinnamic acid [58]. The main identified compound in propolis from Spain, Italy, and India is caffeic acid phenethyl ester (CAPE), an essential polyphenol with mainly anticancer activity [54,59,60,61,62]. Regarding the chemical characterisation of the propolis samples from Romania, Gatea et al. (2015) demonstrated that all propolis samples collected from the central, southern, or western parts of Romania contained high concentrations of pinocembrin (37.56–87.63 mg/g), galangin (18.40–80.85 mg/g), and CAPE (12.83–48.52 mg/g) as the principal compounds [29]. Mărghitaș et al. (2014), by analysing propolis collected from the Transilvania region, found that in all samples, the main compound was chrysin (2.04–3.91 mg/g), followed by galangin (1.67–2.66 mg/g) and pinocembrin (0.69–2.08 mg/kg). Of the polyphenolic acids, the most abundant was CAPE (3.87–0.87 mg/g). Cafeic acid, coumaric acid, or ferulic acid were detected in low concentrations [30]. In contrast to the previous studies from Romania, our studied propolis samples were rich in ferulic acid (126.10–498.72 µg/g) and contained moderate concentrations of rosmarinic acid (68.60–87.50) and coumaric acid (37.00–216.90 µg/g). Even though caffeic acid was detected in a high concentration in CP, it was undetectable in the other two samples. From the flavonoid class, rutin (79.30 µg/g) was noted as the dominant compound, with the concentration of quercetin being the lowest (15.60–76.80 µg/g), except for CP. However, [63] highlighted that the propolis samples collected from five counties in Romania were rich in quercetin (44.65–328.35 µg/mL), which represented the second most abundant flavonoid after kaempferol (475.07–658.94 µg/mL). Thus, the chemical variations of propolis are evident from one region to another. Yet, there was a significant chemical variability between the propolis samples studied in the present study. PC and PT had, as their principal compounds, phenolic acids, ferulic acid (498.72 µg/g), and β-rezorcilic acid, respectively (175.00 µg/g), while MASP was characterised by the predominance of the most well-known among stilbene substances, resveratrol (137.70 µg/g). The second most crucial compound was ferulic acid for MASP, caffeic acid for CP, and resveratrol for PT. This chemical variability can be explained by the different origins of the samples and by various extraction methods [64], and it imprints varied biological activity to the product through the activity of the compounds themselves and their synergistic action. According to the literature, phenolic acids are known for their antidiabetic, anticancer, anti-inflammatory, antimicrobial, antioxidant, and neuroprotective effects [65,66]. Similarly, rutin has various pharmacological applications due to its different biological activities, such as anticancer, antioxidant, neuroprotective, anti-inflammatory, antimicrobial, antidiabetic, hepatoprotective, cardioprotective, and hepatoprotective properties [67].

Unlike propolis, our study showed that the royal jelly contained ferulic acid (127.80–129.60 µg/g) and resveratrol (119.10–233.30 µg/g) as the main compounds. From the flavonoids class, epicatechin was the most abundant substance (73.80–74.20 µg/g), while rutin was undetectable. Like propolis, the concentration of quercetin in all royal jelly samples was low (15.70–15.90 µg/g). The chemical variations from one sample to another were not noticeable. In contrast, Altun et al. (2022) showed significant variations in the chemical composition of commercial royal jelly mixtures. Moreover, the author noted that the chemical composition was characterised by the predominance of chrysin (246.5–5501.1 µg/L) and caffeic acid phenethyl ester (165.1–2431.6 µg/L), while ferulic acid epicatechin, even detected in moderate concentrations, was not the main compound. However, similar to our study, quercetin was detected in low concentrations [68]. These differences regarding the chemical composition of royal jelly are easy to understand when samples from different regions of the world are characterised. Although the physicochemical parameters of this product are frequently determined in the specialised literature [9,69,70], there are no other data on the polyphenolic profile of royal jelly from Romania.

Few studies have investigated the chemical composition of apilarnil, demonstrating that its complex biological activity is due to its chemical composition consisting of fatty acids, flavonoids, glycerophospholipids, alcohols, sugars, amino acids, and steroids. Aida et al. (2024) demonstrated a high number of compounds in apilarnil, specifically, 44. The 30 identified volatile compounds were distinguished as esters, ketones, ethers, alcohols, fatty acids, aldehydes, amines, and alkene [71]. The lipid composition identified by GC/MS consisted of oleic acid (64.75%) and palmitic acid (26.08%), the dominant lipid compounds of apilarnil. Koşum et al. (2022), analysing the phenolic/organic acid profile, observed that trans-aconitic acid (11.20 ± 0.32 μg/g) and fumaric acid (5.03 ± 0.41 μg/g) represented the significant compounds [72]. Inci et al. (2023) showed that the highest compound concentrations, expressed as mg analyte/g dry extract, were represented by quinic acid (1091.04), fumaric acid (48.714), kaempferol (39.946), and quercetin (27.508) [73]. In contrast, in the present study, it could be observed that ferulic acid had the highest amount, 126.36 µg/g, followed by epicatechin with a concentration of 74.03 ± 1.85 µg/g. No gallic acid was determined in all studied extracts. These are the most relevant findings of the present study, as no other Romanian apilarnil types have been determined.

### 4.2. Antimicrobial Activity

The antimicrobial activity of bee products depends on several factors, including chemical composition. From the chemical composition, flavonoids like rutin, apigenin, apigetrin, and astragalin interact with bacteria membranes. The hypothesis is sustained by identifying the release of protein and nucleic acid [74,75,76]. Similarly, phenolic acid causes irreversible damage to the bacteria by its modification of the charge, intra and extracellular permeability, and physicochemical properties of membranes [77]. Furthermore, bee products can have variable bacterial efficacy, depending on the strain studied. For example, [11] demonstrated that all of the propolis extracts evaluated, and collected from different European regions, showed an antibacterial effect against Gram-positive bacterial pathogens, but moderate or no efficacy against Gram-negative bacteria [11]. A low antimicrobial activity of propolis against Gram-negative bacteria was observed in the present study as well. The difference in cell envelopes can explain the variability of bacterial sensitivity to bee products. A Gram-negative bacteria cell envelope consists of an inner cytoplasmic membrane surrounded by a thin peptidoglycan layer (PG) and a lipopolysaccharide outer membrane. Instead, Gram-positive bacteria contain only the PG layers thicker than those observed in Gram-negative bacteria [78].

Similarly, royal jelly was efficient against a broad spectrum of Gram-positive bacteria at different concentrations, while Gram-negative bacteria and fungi registered weak activity [22]. On the other hand, it exhibited higher antibacterial activity against anaerobic bacteria than aerobic ones [23]. However, except for *P. aeruginosa*, most Gram-positive and Gram-negative strains were sensitive to the same concentrations of royal jelly tested.

The MIC values were different even in the same Gram category of bacteria, as the study demonstrated. A comparative study revealed that German, Irish, and Czech propolis had MIC values for *Staphylococcus* spp. between 0.08 and 1.2 mg/mL, mentioning that the lowest MIC value was recorded by using Irish propolis against *S. aureus*. MIC for *Streptococcus* spp. had values similar to those for staphylococci, with the minimum value of 0.08 mg/mL being attributed to Irish and Czech propolis against *S. pneumoniae* and *S. thermophilus* [11]. In contrast, our study showed no antimicrobial activity of the tested propolis against *S. pyogenes* and MIC values of 1–10 mg/mL of the micro apiary sample of propolis against *S. aureus*.

Similarly, Degy et al. (2022) demonstrated that Romanian ethanolic propolis had antimicrobial activity against *S. aureus* with MIC values between 6 and 10 µg/mL [44]. According to the literature, the antimicrobial activity of propolis collected from three European countries against Gram-negative strains was moderate, with MIC values between 0.6 and 5 mg/mL for *E. coli* and *P. aeruginosa*; the Irish propolis was the most efficient [11]. In contrast, the MIC value against *E. coli* strains for the propolis samples tested in the present study was 0.25–1 mg/mL, and it should be mentioned that it did not show efficacy against two pathogenic isolates of the same bacterial species. The MIC values for all propolis samples tested against *P. aeruginosa* were 0.25 mg/mL for two of the pathogenic isolates tested, while the growing of one isolate was not inhibited in the presence of this bee product. Only the micro apiary samples of propolis and the propolis tincture were efficient against the *S. typhimurium* ATCC strain tested, with a MIC value of 0.25 mg/mL. However, according to data from the literature, compared to other *Salmonella* spp., the MIC value of propolis was greater than 5 mg/mL [11].

Royal jelly is another bee product that performs various biological activities. It is known for its beneficial effects on the skin and has demonstrated its antimicrobial effectiveness in numerous studies. Uthaibutra et al. (2023) showed that royal jelly has antimicrobial activity against the most essential skin pathogenic bacteria, with MIC and MBC values ranging from 18.75 to 150.00 mg/mL. Among all the bacterial species studied, the lowest MIC (18.75 mg/mL) value was highlighted in *Corynebacterium* spp. In the case of the *S. aureus* and *S. epidermidis* strains, the authors demonstrated that the MIC value was between 37.50 and 75.00 mg/mL depending on the royal jelly taken into the study on the chemical composition [79]. Instead, the present study showed that MIC values against *S. aureus* were included in the 0.25–1 mg/mL interval. The values are somewhat close to those obtained against most Gram-negative bacteria, except for *P. aeruginosa*. Other authors also described the low efficacity of royal jelly against *P. aeruginosa*. Amly et al. (2021) showed that the pyoceanic pigment was eliminated by using a concentration of 25% of royal jelly [80]. In contrast, [81] showed that the MIC values for *P. aeruginosa* were higher, respectively, 60–100% (*w/w*), but similar to those against *E. coli*. Our study demonstrated that the MIC values of Romanian royal jelly against *P. aeruginosa* were between 0.25 and 5 mg/mL, depending on the type of sample.

Findings on apilarnil suggested that this bee product has various therapeutic activities such as estrogenic and androgenic effects, protecting against testicular damage, protecting against testicular toxicity and liver injury antioxidant capacity, and stimulating the immune system [26,82]. However, the antimicrobial activity was not tested. The present study demonstrated that commercial lyophilised apilarnil had no antimicrobial activity against the ATCC-tested strains but was efficient against *P. aeruginosa* and *E. coli* isolates, with values of MIC of 0.25 mg/mL, respectively, 0.25–0.75 mg/mL. Apilarnil from the micro apiary source and commercial apilarnil demonstrated antimicrobial activity of 0.25 mg/mL against various bacteria strains such as *S. aureus*, *S. flexneri*, *E. coli*, *S. typhimurium*, *H. influenzae*, *B. cereus*, *Cl. perfringens*, and *L. monocytogenes.*

The antifungal activity of various bee products is variable, depending on the same factors as those involved in the antibacterial activity. The mechanism of action consists of the loss of the integrity of the cell wall and increased permeability without mutagenic effects [83] due to the synergistic action of phenolic acids and flavonoids. Among flavonoids, kaempferol, quercetin, and myricetin can inhibit the growth and cell division of *C. albicans* [84]. Moreover, ferulic and gallic acids change cell surface hydrophobicity and charge, while caffeic acid may interfere with 1,3-β-glucan synthase [85]. By comparing some products, studies demonstrated that propolis is the most efficient against *Candida* spp., followed by honey and royal jelly [86].

In contrast, our study demonstrated that royal jelly was efficient against *Candida* spp., while propolis had no activity. The MIC values differed depending on the chemical composition of the bee product and were <5% *v/v* against *C. albicans* and *C. tropicalis* in the case of propolis [87]. The present study showed that only royal jelly had antifungal activity against *C. albicans* and *C. parapsilosis*, with MIC values between 2.5 and 7.5 mg/mL. At the same time, apilarnil and propolis cannot be recommended for treating mycotic skin diseases caused by *C. albicans*. In contrast, Iranian royal jelly and propolis are effective against *C. albicans* [88].

### 4.3. Inhibition of Haemolysis Values and Protein Denaturation

Natural agents that assure lysosomal membrane stabilisation and inhibit protein denaturation could be of great interest in developing anti-inflammatory drugs. By stabilising the lysosomal membrane, the release of chemical mediators involved in the activation of neutrophils is inhibited. Denaturation of tissue proteins is also a cause of inflammatory or arthritic diseases, being associated with the loss of the protein’s biological functions and production of autoantigens [33,89]. Generally, bee products are well-known for their anti-inflammatory activity, even though the mechanism of action is unclear [90,91]. The compounds’ chemical composition, concentration, and synergistic action seem responsible for this biological activity. The presence of rutin in natural compounds gives the products an anti-inflammatory effect by reducing the effects on the level of NO or PGE2 and pro-inflammatory cytokines TNF-α and IL-6 [92]. Moreover, this flavonoid acts synergistically with quercetin on inflammatory markers (CRP, IL-1β, and IL-6) and various oxidative markers [93]. Similarly, ferulic acids inhibit the activation of NLRP3 inflammasome and reduce the expression and release of inflammatory factors [94]. The protective activity of red blood cell membranes from oxidative injuries is attributed to ferulic and caffeic acids [95,96]. Regarding the inhibition of haemolysis values and protein denaturation results, our study demonstrated that propolis had a protective effect against RBC damage and protein denaturation starting with a concentration of 2.5%, except for the propolis tincture. Increasing membrane stability is due to ferulic and caffeic acids and flavonoids [96,97]. Moreover, ferulic acid and flavonoids also seem to have a protective effect against protein denaturation by inhibiting the loss of their primary, secondary, and tertiary structures [98,99,100]. However, the mechanism of protection is a more complex process, which is based on interactions between the components of propolis. Other authors also demonstrated the capacity of propolis to protect against haemolysis and protein denaturation. Mendez-Encinas et al. (2023) showed the fact that seasonal extracts of Sonoran propolis had effects on RBC damage depending on the time of collection and the dose, as well as the fact that variations directly influenced their chemical composition [89]. Other research by [101] demonstrated similar effects to our study using Korean propolis, extracted with ethanol and used as a test material by oral administration (100 mg/kg). The results demonstrated the significant inhibition of the increase in vascular permeability and of acetic acid-induced writhing in mice. Thus, Korean propolis was shown to have intense RBC protective activity. As far as we know, there are no data in the literature on the protective activity against the protein denaturation or red cell damage of royal jelly and apilarnil. However, our study demonstrated that even though both bee products had positive values of the inhibition of the protein denaturation, they did not protect red cells from heat-induced haemolysis. The exception was represented by commercial lyophilised royal jelly, which demonstrated a protective effect against RBC damage starting with a concentration of 2.5%.

### 4.4. Docking Analysis

Using an in silico model to describe the biological activities of compounds identified from natural products has recently gained appreciable attention [102]. This has further improved the knowledge of possible and predictable interactions between the compounds of natural products and the studied crystallized protein structures of microorganisms, improving the understanding of antimicrobial relevance. The current study sought to adopt this model in defining the antimicrobial activities of bee products with visible modes of interaction between the compounds identified by the LC-MS of the bee products and protein structures of bacteria (1JII, 1G27, 3LPX) and fungi (2QA1, 8JZN, 5AG7).

The result of the docking between tyrosyl-tRNA synthetase and the compounds predicted the strongest affinity for caffeic acid, gallic acid, resveratrol, and rutin, with gallic acid shown to be the most potent bioactive molecule of the bee products against this protein (Table 7 and Figure 1). This is in agreement with the findings by Elkolli et al. (2024) [49], wherein the authors of the study showed that the bioactive compound from their study interacted with tyrosyl-tRNA synthetase residues CYP37, LEU70, THR75, ASN124, TYR170, and ILE200. Interestingly, caffeic acid, gallic acid, resveratrol, and rutin would have had this interaction because of the free hydroxyl groups in their structures, allowing for stronger affinity and binding to this protein.

The interaction profile of the compounds docked with peptide deformylase showed an excellent fit of epicatechin, ferulic acid, gallic acid, and rosmarinic acid in the binding pocket of the protein, GLN50, GLU133, GLY81, LEU91, GLN65, LEU112, VAL50, and GLY110 (Figure 2). A similar observation was made by Merzoug et al. (2017) [103]. Other studies by [104,105] found compounds with hydroxamate functional groups to be viable antibacterial agents when targeting the peptide deformylase protein.

The entire interaction of the compounds with DNA gyrase was more plausible with rosmarinic acid and rutin, which had a binding energy of −7.6 kcal/mol and 9.0 kcal/mol, respectively. This is owing to the stronger interaction of the protein with aspartate, histidine, threonine, asparagine, and glutamine amino acids, which could lead to DNA breakage and disallow the ligating opportunity of the protein. Studies by [106,107] support this assertion.

The results of the protein–ligand interaction between the compounds and fungi proteins indicate high binding energies for squalene epoxidase and epicatechin (−8.0 kcal/mol), gallic acid (−6.8 kcal/mol), and rosmarinic acid (−8.1 kcal/mol), wherein the strongest interaction could be seen between the hydroxyl group of these compounds and 2QA1 amino acid residues GLU32, LEU31, VAL120, MET288, ALA254, GLN96, and ARG42 (Figure 4). The different hydrogen bonds and other pi-hydrophobic interactions are similar to those of Nowosielski et al. [38], where the authors docked terbinafine against squalene epoxidase. For the interaction between 1,3-β-glucan synthase and the compounds, rutin (−9.4 kcal/mol) showed the formation of more hydrogen bonds, carbon-hydrogen bonds, and few hydrophobic pi-bonds (PRO847, LYS1082, ASP1102, GLN1214; GLU851, ASP1080; TYR849, ASP1222). Rutin as a flavonoid was previously reported to have antifungal activity against some fungal strains and was suggested to be chemically modified when there was the introduction of a substitute that could change the physicochemical properties of rutin and thus increase the antifungal activity [108]. N-myristoyl transferase showed that gallic acid and rutin had the most interesting interactions, seemingly due to the mentioned properties of these compounds in potentiating their activities against fungal proteins. These compounds had a high affinity to the protein’s binding pocket and implicated both aliphatic and aromatic amino acids in the binding. This agrees with the report of [109] where the authors demonstrated this class of amino acids in the binding affinity of a sulphonamide with N-myristoyl transferase.

Cumulatively, the antimicrobial potential of the bee products is, no doubt, exercised due to the presence of these compounds to varying degrees. Hence, explaining their activities on the studied protein targets and rendering them significant to further inhibition usage are paramount.

All the data suggest that finding alternative methods of treatment in mycotic and bacterial diseases is a long process that requires in-depth studies due to the variations in the biological effects of natural products depending on their chemical composition.

## 5. Conclusions

In conclusion, the chemical composition of bee products such as propolis, royal jelly, and apilarnil varies significantly depending on their origin and extraction methods. This variability directly impacts their biological activities, including antimicrobial properties. The antimicrobial activity of bee products showed specificity towards certain bacterial strains. Propolis demonstrated efficacy primarily against Gram-positive bacteria, while royal jelly exhibited moderate activity against Gram-positive and Gram-negative bacteria. Apilarnin proved to have significant inhibitory results, promoting it as a potential antimicrobial agent in future research. The MIC values varied, reflecting differences in bacterial strains and bee products. Notably, Romanian propolis showed promising antibacterial activity against *S. aureus*. Furthermore, our study highlighted the potential protective effects of propolis, suggesting its role in protecting against acute and chronic inflammation. Molecular docking studies revealed potential interactions between phenolic compounds found in bee products and various bacterial and fungal proteins, indicating their mechanisms of action.

Overall, these findings underscore the importance of understanding bee products’ chemical composition and biological activities for their potential therapeutic applications, while emphasising the need for further research to elucidate their mechanisms of action and optimise their use in healthcare.

## Figures and Tables

**Figure 1 foods-13-01455-f001:**
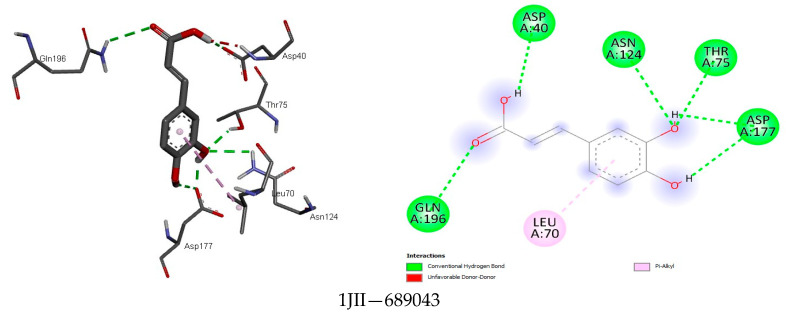
The binding mode of some identified compounds in Tyrosyl-tRNA synthetase shows strong interactions with key amino acids.

**Figure 2 foods-13-01455-f002:**
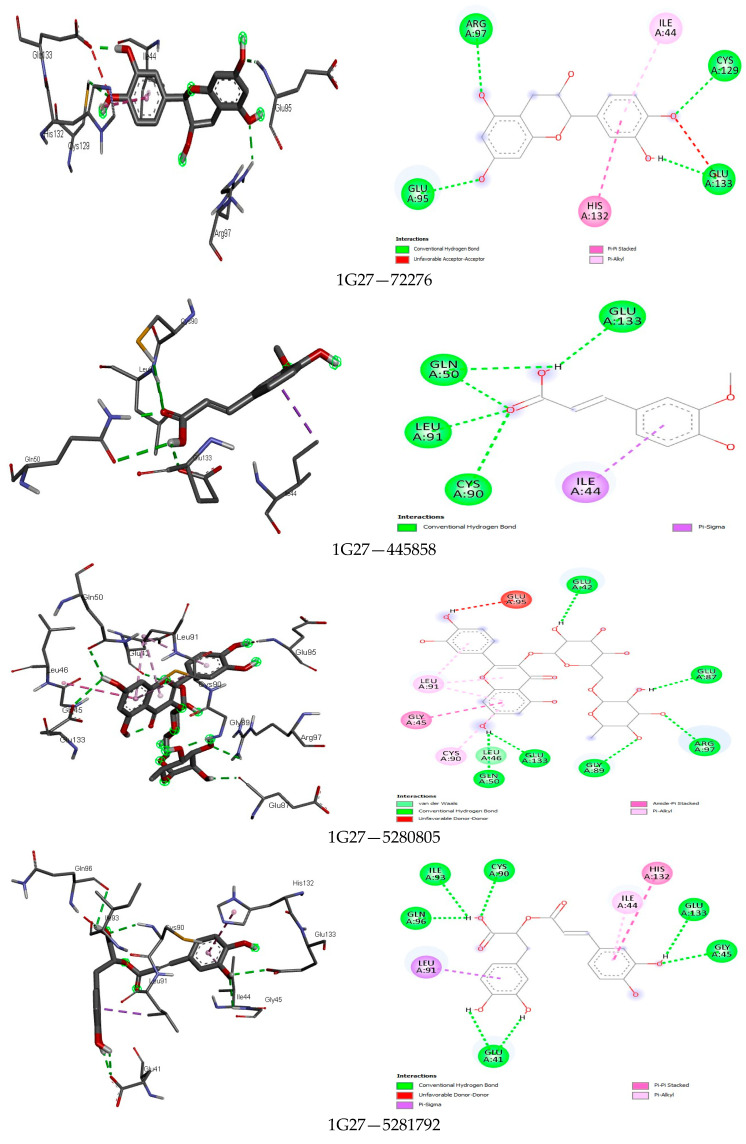
The binding mode of some identified compounds in peptide deformylase shows strong interactions with key amino acids.

**Figure 3 foods-13-01455-f003:**
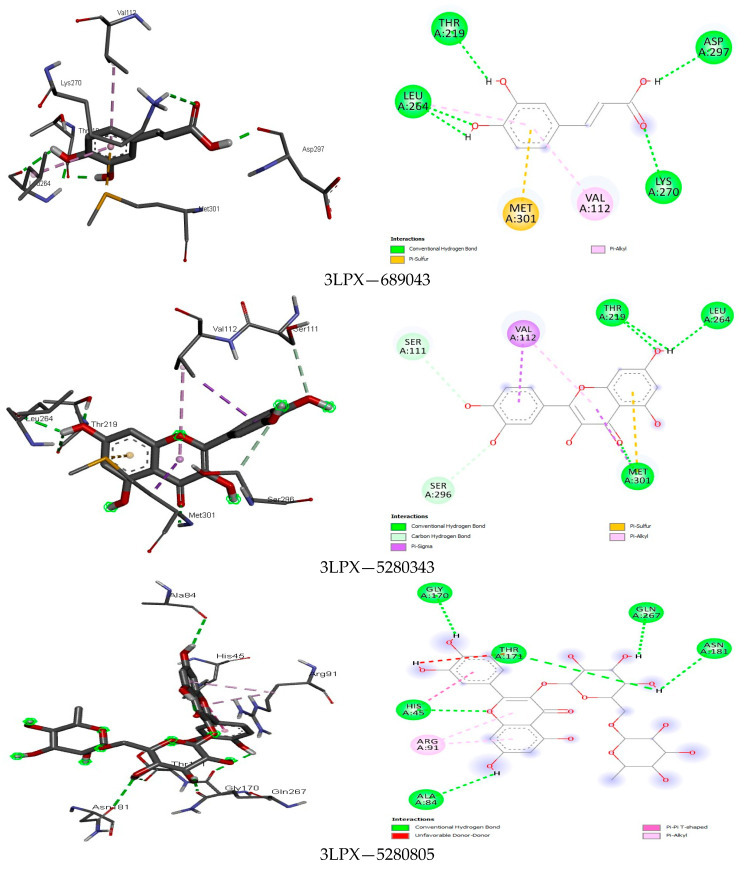
The binding mode of some identified compounds in DNA gyrase shows strong interactions with key amino acids.

**Figure 4 foods-13-01455-f004:**
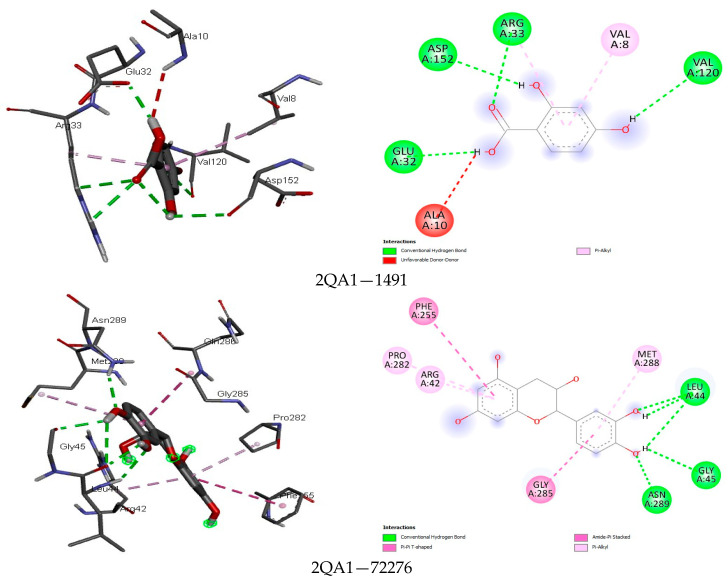
The binding mode of some identified compounds in squalene epoxidase shows strong interactions with key amino acids.

**Figure 5 foods-13-01455-f005:**
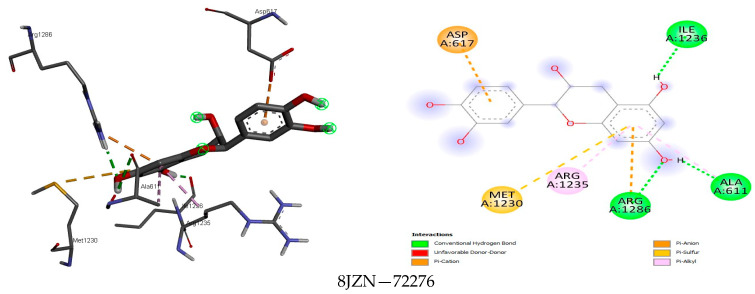
The binding mode of some identified compounds in 1,3-β-glucan synthase shows strong interactions with key amino acids.

**Figure 6 foods-13-01455-f006:**
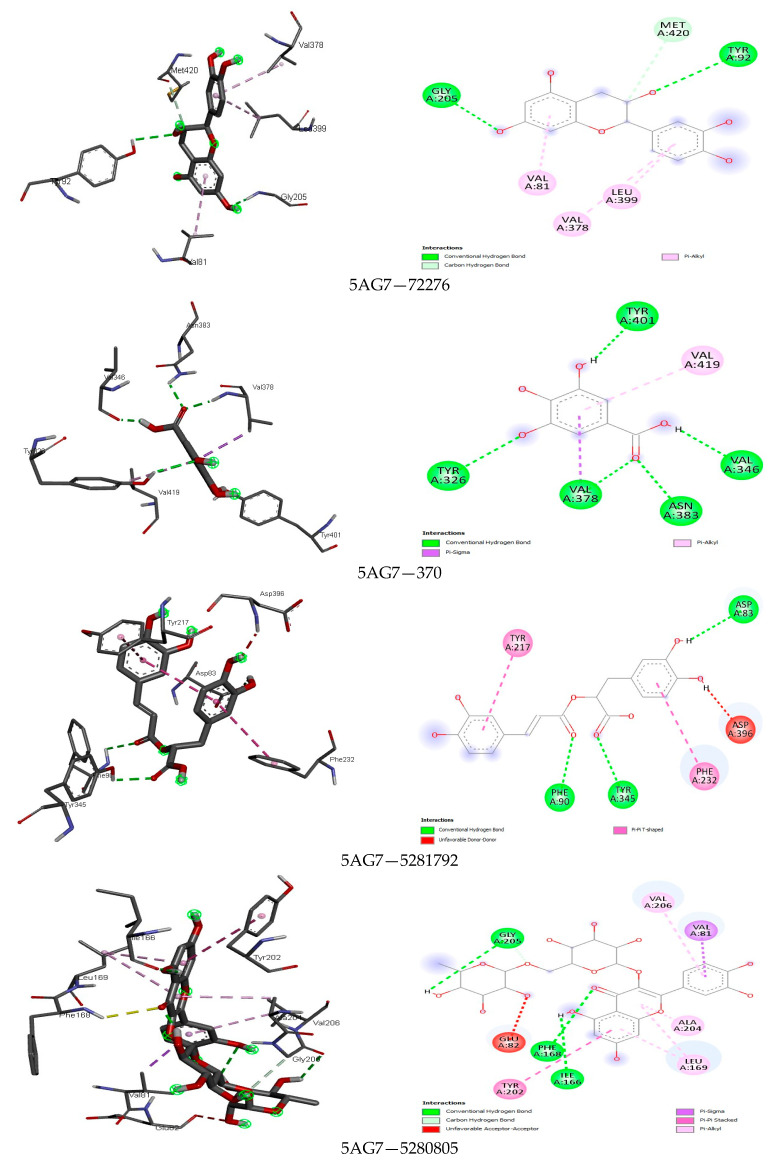
The binding mode of some identified compounds into N-myristoyl transferase shows strong interactions with key amino acids.

**Table 1 foods-13-01455-t001:** Individual polyphenols (mg/g) identified using LC/MS.

Class	Compound	MASP	CP	PT	MASRJ	CLRJ	CLJ	MASA	CLA	CA
Flavonoids	Epicatechin (µg/g d.w.)	73.90 ± 1.20 ^a^	73.90 ± 1.85 ^a^	74.20 ± 1.86 ^a^	73.80 ± 1.85 ^a^	74.20 ± 1.86 ^a^	74.00 ± 1.85 ^a^	73.90 ± 1.85 ^a^	74.20 ± 1.86 ^a^	74.00 ± 1.85 ^a^
	Rutin (µg/g d.w.)	79.30 ± 1.98 ^a^	79.30 ± 1.98 ^a^	79.30 ± 1.98 ^a^	nd *	nd	nd	79.30 ± 1.98 ^a^	79.30 ± 1.98 ^a^	79.00 ± 4.38 ^a^
	Quercitin (µg/g d.w.)	15.60 ± 0.39 ^a^	16.80 ± 1.92 ^c^	15.70 ± 0.39 ^a^	15.78 ± 0.39 ^a^	15.90 ± 0.40 ^a,b^	15.70 ± 0.39 ^a^	16.10 ± 0.40 ^b^	16.50 ± 0.41 ^b,c^	15.80 ± 0.40 ^a,b^
	Total (µg/g d.w.)	168.8 ± 3.57	170 ± 5.75	169.2 ± 5.76	89.58 ± 2.24	90.1 ± 2.26	89.7 ± 2.24	169.3 ± 4.23	170 ± 4.25	168.8 ± 6.63
Phenolic acids	Gallic acid (µ/gd.w.)	17.70 ± 0.44 ^b^	16.90 ± 0.42 ^a^	16.90 ± 0.42	nd	nd	nd	nd	nd	nd
	β-rezolcilic acid (µg/g d.w.)	nd	nd	nd	nd	175.20 ± 4.38 ^a^	175.00 ± 4.38 ^a^	175.10 ± 4.38 ^a^	175.10 ± 4.38 ^a^	175.00 ± 4.38 ^a^
	Caffeic acid (µg/g d.w.)	nd	nd	nd	nd	nd	nd	18.30 ± 0.46 ^a^	19.40 ± 0.49 ^b^	19.30 ± 0.38 ^a^
	Cumaric acid (µg/g d.w.)	41.00 ± 1.03 ^c^	50.90 ± 5.42 ^d^	37.00 ± 0.93 ^a^	39.60 ± 0.99 ^b^	39.60 ± 0.99 ^b^	39.10 ± 0.98 ^b^	39.80 ± 0.02 ^b^	37.70 ± 0.94 ^a^	40.80 ± 1.02 ^c^
	Ferulic acid (µg/g d.w.)	126.40 ± 3.16 ^a^	198.72 ± 12.47 ^c^	127.00 ± 3.18 ^a^	129.60 ± 3.24 ^b^	126.60 ± 3.17 ^a^	127.80 ± 3.20 ^a^	126.80 ± 3.17 ^a^	126.10 ± 3.15 ^a^	126.20 ± 3.16 ^a^
	Rosmarinic acid (µg/g d.w.)	67.90 ± 1.70 ^a^	68.60 ± 1.72 ^a,b^	87.50 ± 2.19 ^c^	68.10 ± 1.70 ^a,b^	69.10 ± 1.73 ^b^	68.20 ± 1.71 ^a,b^	67.90 ± 1.70 ^a^	68.00 ± 1.70 ^a^	68.00 ± 1.70 ^a^
	Total (µg/g d.w.)	253 ± 6.33	335.2 ± 20.03	268.4 ± 6.72	237.3 ± 5.93	410.5 ± 10.27	410.1 ± 10.27	427.9 ± 9.73	426.3 ± 10.66	429.3 ± 10.64
Stilbene	Resveratrol (µg/g d.w.)	137.70 ± 3.44 ^c^	122.60 ± 3.07 ^a^	132.70 ± 3.32 ^b,c^	119.10 ± 2.98 ^a^	233.30 ± 5.83 ^e^	226.60 ± 5.67 ^e^	135.00 ± 4.38 ^a^	128.20 ± 3.21 ^b^	160.00 ± 4.00 ^d^
	Total (µg/g d.w.)	137.70 ± 3.44 ^c^	122.60 ± 3.07 ^a^	132.70 ± 3.32 ^b,c^	119.10 ± 2.98 ^a^	233.30 ± 5.83 ^e^	226.60 ± 5.67 ^e^	135.00 ± 4.38 ^a^	128.20 ± 3.21 ^b^	160.00 ± 4.00 ^d^

Results are expressed as the three-determination mean ± standard deviation (SD). Different lowercase letters (a–e) indicate statistically significant differences between samples according to the *t*-test; d.w.—dry weight; * nd—not detectable.

**Table 2 foods-13-01455-t002:** Bacterial inhibitory rate (% of Gram-positive ATCC strains).

Sample mg/mL/Strain	*S. pyogenes* ATCC 19615	*S. aureus* ATCC 25923	*B. cereus* ATCC 10876	*Cl perfringens* ATCC 13124	*L. monocytogenes* ATCC 19114
MASP 2.5	−126.65	30.25	−78.55	−22.46	−13.61
MASP 5	−146.08	19.61	−51.04	−31.69	1.29
MASP 7.5	−147.75	13.12	−19.9	−42.26	7.9
MASP 10	−149.53	4.42	−19.15	−43.38	13.94
MASP 25	−163.22	−0.09	−16.32	−137.44	17.44
MASP 50	−168.76	−11.88	−12.92	−113.74	28.87
MASP 75	−196.87	−20.49	−11.42	−176.92	34.26
MASP 100	−210.97	−47.93	18.51	−224.31	36.04
CP 2.5	−80.36	55.71	40.02	12.82	34.95
CP 5	−90.07	49.86	32.41	−1.64	37.1
CP 7.5	−96.03	46.87	16.32	−32.62	39.84
CP 10	−102.82	45.95	11.3	−37.03	41.1
CP 25	−106.58	27.49	11.13	−68.31	41.46
CP 50	−111.6	18.92	10.03	−70.46	43.87
CP 75	−135.21	17.13	9.46	−73.74	46.68
CP 100	−154.02	13.95	4.61	−91.28	51.04
PT 2.5	−86.52	−30.76	−68.22	−58.87	−94.35
PT 5	−115.78	−32	−88	−61.03	−76.71
PT 7.5	−163.11	−37.71	−91.41	−104.41	−69.08
PT 10	−172.73	−48.76	−94	−131.49	−14.97
PT 25	−327.17	−118.19	−104.33	−264.92	−12.39
PT 50	−409.2	−134.48	−106.75	−353.95	−2.48
PT 75	−441.8	−139.18	−107.84	−472.21	2.84
PT 100	−482.65	−143.42	−150.52	−534.97	14.17
MASRJ 2.5	−33.75	22.79	7.15	29.64	39.18
MASRJ 5	−23.3	38.86	16.44	39.08	45.13
MASRJ 7.5	−16.82	49.68	18.28	43.59	48.5
MASRJ 10	−14.42	59.39	36.79	47.79	57.05
MASRJ 25	12.54	57.46	47.69	47.9	62.14
MASRJ 50	38.35	68.78	73.47	54.77	80.64
MASRJ 75	42.11	70.3	76.93	55.59	84.01
MASRJ 100	45.98	72.1	78.03	60.62	84.18
CRJ 2.5	−253.5	−23.02	−5.88	−95.38	24.88
CRJ 5	−209.2	−11.23	3.86	−50.05	33.27
CRJ 7.5	−163.85	−7.46	15.74	−4.41	43.84
CRJ 10	−145.35	1.2	30.1	23.08	56.16
CRJ 25	−127.59	12.29	37.37	51.9	60.92
CRJ 50	21.32	53.18	73.36	61.95	82.72
CRJ 75	44.51	78.36	74.74	61.74	83.98
CRJ 100	48.69	78.91	75.78	62.77	84.41
CLRJ 2.5	−167.71	41.3	−39.45	−172.72	11.63
CLRJ 5	−138.24	50.83	−30.57	−150.77	17.91
CLRJ 7.5	−107.63	57.83	−17.07	−133.85	21.18
CLRJ 10	−33.33	64.55	−8.3	−105.13	28.15
CLRJ 25	−15.88	76.93	−0.63	−61.95	34.75
CLRJ 50	38.24	77.81	51.5	32.92	66.3
CLRJ 75	41.69	78.31	71.97	56.41	84.64
CLRJ 100	47.65	79.01	73.93	55.9	84.7
MASA 2.5	−64.79	53.22	10.9	29.54	67.16
MASA 5	−65.41	51.1	6.11	25.85	64.95
MASA 7.5	−89.24	14.83	−16.03	15.18	47.08
MASA 10	−92.37	−5.48	−21.28	11.08	11.3
MASA 25	−204.28	−9.25	−72.84	−128.62	9.32
MASA 50	−345.66	−93.69	−163.55	−332.31	−45.66
MASA 75	−465.31	−151.93	−220.76	−469.85	−90.98
MASA 100	−665.41	−241.8	−323.7	−630.97	−144.33
CLA 2.5	−24.03	56.81	20.82	57.23	51.17
CLA 5	−17.55	63.81	21.63	56.31	66.9
CLA 7.5	−17.14	64.5	44	56.21	66.93
CLA 10	−16.93	65.84	44.41	50.15	65.94
CLA 25	−11.39	66.3	61.36	46.36	72.91
CLA 50	21.84	67.91	60.73	42.36	78.72
CLA 75	20.48	70.17	68.74	40.62	79.19
CLA 100	15.36	76.29	68.8	35.9	80.38
CA 2.5	−36.68	77.95	19.26	40.92	66.73
CA 5	−42.84	76.43	18.34	40.62	66.07
CA 7.5	−56.11	74.82	−0.81	19.18	58.57
CA 10	−88.09	54.33	−20.07	5.85	56.56
CA 25	−115.46	37.57	−23.76	−36.51	24.61
CA 50	−142.22	−10.82	−44.29	−150.87	24.25
CA 75	−251.72	−62.02	−110.32	−282.26	−7.83
CA 100	−360.5	−120.26	−176.99	−393.85	−56.92

**Table 3 foods-13-01455-t003:** BIR% of Gram-negative bacteria and fungal *Candida* ATCC strains.

Sample/Strain	*S. flexneri*	*P. aeruginosa*	*E. coli*	*S. typhimurium*	*H. influenzae*	*C. parapsilopsis*	*C. albicans*
MASP 2.5	131F.06	−14.03	−41.01	50.17	24.49	−45.08	−128.66
MASP 5	123.78	−18.21	−34.07	50.07	16.95	−22.53	−232.11
MASP 7.5	112.64	−34.15	−9.28	41.61	15.02	−19.56	−360.16
MASP 10	107.68	−62.57	0.9	34.08	11.95	−7.51	−360.57
MASP 25	106.26	−89.25	6.37	32.18	5.91	0.88	−397.56
MASP 50	106.22	−138.43	8.11	29.64	−4.24	14.51	−456.71
MASP 75	96.79	−138.89	17.39	25.78	−6.27	14.81	−525.2
MASP 100	84.51	−171.95	24.19	12.96	−14.41	16.16	−555.08
CP 2.5	90.08	1.46	30.15	53.95	28.43	29.39	−314.43
CP 5	83.21	−13.11	31.69	56.09	28.66	29.97	−249.8
CP 7.5	82.56	−21.4	35.41	56.12	28.63	31.28	−228.66
CP 10	79.31	−32.79	36.68	56.33	32.8	31.92	−225.61
CP 25	79.02	−43.9	38.79	56.52	37.67	33	−203.66
CP 50	70.16	−52.46	41.17	57.7	40.87	33.23	−200.61
CP 75	60.49	−58.47	48.78	57.94	53.22	39.87	−188.62
CP 100	58.25	−64.94	52.86	60	62.66	42.22	−156.3
PT 2.5	105.16	−41.62	12.53	41.02	−15.05	−76.73	−213.01
PT 5	109.23	−49.64	10.22	37.8	−15.62	−72.96	−329.07
PT 7.5	126.1	−76.23	9.18	13.9	−15.75	−68.72	−354.27
PT 10	129.55	−139.16	−19.43	10.15	−18.32	−62.86	−482.32
PT 25	133.17	−244.08	−80.23	8.25	−35.84	−16.16	−567.07
PT 50	182.15	−335.15	−80.5	−20.83	−67.37	−15.49	−778.25
PT 75	193.29	−371.31	−81.47	−29.93	−69.47	−13.77	−845.33
PT 100	200.69	−394.9	−89.45	−30.44	−78.68	−2.73	−871.54
MASRJ 2.5	79.19	21.86	30.28	51.49	25.66	25.35	−133.74
MASRJ 5	70.08	19.4	35.08	56.92	25.99	30.27	−116.46
MASRJ 7.5	64.72	15.76	50.25	61.23	28.03	33.64	−110.98
MASRJ 10	60.81	12.57	62.75	61.5	29.5	33.74	−108.74
MASRJ 25	57.6	8.74	67.47	86.02	53.82	54.81	−60.77
MASRJ 50	40.73	2.82	76.92	87.66	79.05	82.42	−11.59
MASRJ 75	23.86	−5.65	80.7	88.35	81.65	83.43	1.42
MASRJ 100	20.98	−4.83	85.66	89.05	83.38	85.02	14.63
CRJ 2.5	81.1	−33.06	49.82	50.15	19.95	14.92	−188.21
CRJ 5	66.5	−9.84	57.39	52.9	28.5	20.74	−142.68
CRJ 7.5	59.72	8.65	61.98	58.8	37.54	29.09	−86.18
CRJ 10	50.39	20.49	69.48	71.11	43.64	63.7	−60.77
CRJ 25	34.27	39.71	77.52	85.25	51.15	46.67	6.1
CRJ 50	18.29	60.29	84.69	88.49	69.7	85.89	18.7
CRJ 75	16.99	62.02	85.73	88.92	82.08	86.9	33.94
CRJ 100	16.38	62.3	86.93	88.81	84.25	87.91	23.98
CLRJ 2.5	104.11	−116.67	13.94	48.03	11.34	10.57	−381.5
CLRJ 5	96.95	−92.81	21.11	53.95	14.61	19.97	−326.83
CLRJ 7.5	88.27	−70.31	28.31	64.79	20.25	27.74	−291.67
CLRJ 10	77.44	−41.35	37.29	72.96	27.19	35.82	−239.84
CLRJ 25	62.8	−18.12	44.72	81.07	38.61	40.37	−178.25
CLRJ 50	33.46	56.65	83.32	87.6	63.56	83.74	−76.02
CLRJ 75	18.86	57.83	84.25	87.87	82.68	84.61	17.89
CLRJ 100	18.9	58.74	85.13	88.35	84.15	90.51	23.37
MASA 2.5	46.59	28.69	58.93	60.29	59.29	55.39	−104.67
MASA 5	47.03	13.3	58.36	56.97	58.99	44.28	−125.81
MASA 7.5	49.23	7.92	42.35	52.29	52.49	29.46	−183.74
MASA 10	54.31	−9.84	14	43.43	36.54	12.15	−225.61
MASA 25	93.17	−103.92	6.4	40.11	9.28	11.85	−394.51
MASA 50	170.28	−293.08	−43.85	1.31	−50.12	−26.63	−663.21
MASA 75	231.5	−426.14	−99.43	−35.26	−92.69	−79.12	−1016.06
MASA 100	289.72	−600.55	−148.44	−85.38	−147.35	−142.79	−1415.04
CLA 2.5	56.54	57.83	83.65	61.39	46.28	34.18	−204.88
CLA 5	42.89	51.91	81.44	63.59	49.45	43.7	−165.04
CLA 7.5	39.11	51.46	77.82	68.94	50.15	59.49	−128.46
CLA 10	35.65	48.54	71.29	69.75	52.45	63.94	−85.16
CLA 25	31.91	44.35	67.74	83.56	62.06	76.6	−55.28
CLA 50	31.42	43.99	61.07	86.13	70.64	80.1	−21.34
CLA 75	28.46	38.52	60.44	87.1	73.61	82.79	6.1
CLA 100	24.63	20.49	57.39	89.37	78.28	84.21	16.06
CA 2.5	30.53	29.87	65.09	66.72	44.51	58.28	−189.23
CA 5	43.21	16.12	58.22	61.93	41.14	51.85	−195.53
CA 7.5	44.88	−0.27	51.29	61.69	41.11	38.59	−207.93
CA 10	48.37	−8.65	32.43	54.08	35.1	28.59	−225.2
CA 25	62.24	−60.66	24.09	40.27	23.49	22.96	−307.32
CA 50	92.4	−132.79	15.14	34.56	22.09	15.89	−354.88
CA 75	144.92	−233.06	−16.05	7.6	−17.08	−18.65	−603.25
CA 100	175.85	−341.62	−51.19	−22.78	−51.48	−54.71	−818.09

**Table 4 foods-13-01455-t004:** BIR% of wild isolates.

Sample mg/mL/Strain	*S. aureus* (MLIRP 052022)	*S. aureus* (MLIRP 072022)	*S. aureus* (MLIRP 092020)	*P. aeruginosa* (MLIRP 122021)	*P. aeruginosa* (MLIRP 092022)	*P. aeruginosa* (MLIRP 042019)	*E. coli* (MLIRP 062019)	*E. coli* (MLIRP 022020)	*E. coli* (MLIRP 112020)
MASP 0.25	−17.26	−4.18	−14.97	4.69	0.05	2.05	3.97	−0.52	−3.20
MASP 0.5	−13.71	−5.50	−15.29	1.24	−1.67	1.05	2.94	−1.63	−3.75
MASP 0.75	−11.07	−6.32	−15.08	−5.00	−3.52	−0.08	2.60	−3.47	−4.30
MASP 1.0	−4.97	−8.87	−16.34	−5.67	−7.81	−1.84	2.26	−4.58	−5.04
MASP 2.5	−2.54	−10.91	−18.43	−7.92	−9.95	−2.22	1.23	−4.85	−6.24
MASP 5.0	−0.71	−11.62	−19.69	−10.58	−13.24	−3.73	0.03	−6.42	−6.97
MASP 7.5	−0.10	−12.84	−20.63	−14.17	−15.95	−4.86	−0.74	−6.97	−8.26
MASP 10	1.93	−13.25	−22.30	−16.82	−18.95	−6.88	−1.69	−8.08	−8.91
CP 0.25	−22.94	−2.45	−3.87	1.77	−2.10	0.67	5.17	−0.71	−1.54
CP 0.5	−22.03	−3.06	−5.97	0.44	−2.81	−0.08	4.83	−1.90	−2.09
CP 0.75	−20.61	−6.32	−7.33	−0.62	−3.24	−1.97	3.89	−2.18	−2.83
CP 1.0	−18.38	−8.15	−8.38	−3.81	−4.95	−4.23	3.46	−3.38	−3.20
CP 2.5	−15.74	−9.68	−9.63	−5.40	−7.10	−4.86	2.77	−4.12	−3.66
CP 5.0	−13.50	−10.40	−10.37	−7.66	−7.81	−5.62	2.17	−5.50	−4.12
CP 7.5	−4.47	−11.42	−11.83	−10.18	−10.10	−6.62	1.49	−6.24	−5.41
CP 10	−1.22	−13.05	−13.09	−10.71	−12.38	−9.01	0.03	−7.34	−6.05
PT 0.25	−22.74	−2.14	−5.76	1.51	−1.52	0.29	3.80	−1.08	−2.00
PT 0.5	−20.00	−3.77	−7.23	−0.35	−2.81	−0.46	2.86	−2.09	−2.55
PT 0.75	−19.29	−4.49	−7.64	−1.28	−3.24	−1.84	2.17	−3.75	−3.29
PT 1.0	−17.26	−7.54	−8.38	−3.54	−5.52	−2.47	1.74	−4.30	−4.21
PT 2.5	−14.52	−10.81	−9.42	−4.47	−7.10	−4.36	0.97	−5.04	−4.67
PT 5.0	−12.59	−11.93	−10.79	−5.27	−8.24	−6.88	0.46	−6.42	−5.50
PT 7.5	−11.07	−12.95	−11.83	−8.32	−9.52	−8.13	−0.66	−7.25	−6.33
PT 10	−8.12	−14.78	−12.88	−12.04	−12.67	−9.01	−1.17	−8.17	−6.88
MASRJ 0.25	−0.71	−4.18	−4.92	−1.81	−0.10	0.17	12.55	8.14	−0.80
MASRJ 0.5	−1.32	−2.96	−4.40	−2.88	−1.38	−2.22	11.52	7.31	−1.26
MASRJ 0.75	−5.18	−2.04	−3.87	−4.07	−2.24	−4.36	10.66	7.13	−2.37
MASRJ 1.0	−10.46	−1.33	−3.25	−5.27	−3.95	−6.00	8.69	5.93	−2.73
MASRJ 2.5	−12.49	−0.20	−2.41	−5.67	−4.81	−8.13	8.26	5.10	−3.10
MASRJ 5.0	−14.52	1.53	−1.47	−7.53	−6.81	−9.39	6.98	3.90	−3.84
MASRJ 7.5	−15.84	2.85	2.41	−9.12	−8.52	−10.40	6.46	3.07	−5.04
MASRJ 10	−17.16	3.36	3.87	−12.17	−9.52	−11.03	5.69	2.43	−6.33
CLRJ 0.25	−3.76	−4.18	−3.46	1.37	1.62	10.10	6.38	9.62	−0.98
CLRJ 0.5	−7.01	−3.26	−2.72	−0.35	0.76	9.22	6.03	9.25	−1.63
CLRJ 0.75	−7.61	−2.34	−2.09	−2.08	0.05	8.22	5.60	8.79	−2.18
CLRJ 1.0	−10.46	−1.53	−1.36	−2.61	−0.81	7.71	4.83	8.23	−2.92
CLRJ 2.5	−12.28	−0.51	−0.31	−3.54	−2.38	6.83	3.54	7.68	−3.66
CLRJ 5.0	−12.99	0.31	0.31	−4.47	−3.52	4.57	2.86	6.67	−4.12
CLRJ 7.5	−14.92	1.63	0.73	−5.40	−4.38	3.19	2.17	6.21	−4.76
CLRJ 10	−16.55	2.96	1.26	−6.33	−5.95	0.67	0.97	5.47	−5.78
CRJ 0.25	−0.20	−2.34	−5.55	−5.40	4.48	6.08	13.58	10.08	7.31
CRJ 0.5	−0.81	−1.73	−4.40	−6.20	3.48	4.44	12.46	9.43	6.67
CRJ 0.75	−1.93	−1.12	−3.87	−7.13	1.62	3.19	11.86	8.88	6.11
CRJ 1.0	−8.22	−0.41	−2.93	−7.79	0.90	2.18	11.18	8.23	5.19
CRJ 2.5	−9.95	0.00	−2.09	−10.18	−1.10	1.05	10.75	7.40	3.90
CRJ 5.0	−11.37	0.51	−1.36	−11.51	−2.67	−0.08	10.06	6.85	2.80
CRJ 7.5	−15.23	1.33	−0.31	−12.17	−3.95	−0.96	8.86	6.02	1.78
CRJ 10	−18.27	2.45	0.63	−14.56	−5.67	−2.10	8.18	5.01	0.86
MASA 0.25	−1.62	−0.41	−1.05	−3.54	0.76	2.43	−0.31	7.86	−2.92
MASA 0.5	−1.22	0.31	−0.42	−4.34	0.05	0.67	0.29	8.42	−2.27
MASA 0.75	−0.41	1.02	0.52	−4.87	−1.24	0.42	1.32	8.97	−1.35
MASA 1.0	0.51	1.83	1.26	−5.53	−1.67	−0.96	1.83	9.16	−0.89
MASA 2.5	1.32	2.96	2.09	−6.06	−3.10	−1.97	3.12	9.62	0.31
MASA 5.0	2.23	3.77	3.04	−7.26	−4.10	−2.85	4.23	10.35	1.14
MASA 7.5	2.44	4.59	3.77	−8.46	−4.95	−4.11	5.00	10.63	2.33
MASA 10	3.05	5.30	4.40	−9.25	−6.67	−4.86	6.38	11.46	3.35
CLA 0.25	1.52	0.20	−1.78	2.57	4.76	9.10	5.26	9.16	−1.17
CLA 0.5	2.44	0.51	−1.36	1.51	2.90	8.22	6.46	9.62	−0.43
CLA 0.75	3.35	0.92	−0.63	0.58	1.48	6.96	7.32	10.17	0.40
CLA 1.0	4.26	1.83	0.00	−0.89	0.76	6.21	7.92	10.63	1.04
CLA 2.5	5.58	2.45	1.36	−1.81	−0.38	4.44	9.12	11.37	2.24
CLA 5.0	6.50	3.67	2.30	−3.14	−2.24	3.31	10.06	11.83	3.07
CLA 7.5	6.90	4.69	3.46	−4.74	−2.95	1.93	11.09	12.38	4.18
CLA 10	7.82	5.40	4.19	−5.40	−4.67	0.29	12.21	13.49	5.10
CA 0.25	−2.34	0.71	−2.93	−0.89	−0.81	4.82	−0.40	8.42	−2.09
CA 0.5	−1.52	1.22	−2.20	−1.81	−1.24	3.19	0.37	8.79	−1.44
CA 0.75	−0.81	1.83	−1.47	−1.95	−2.38	2.56	1.74	9.62	−0.89
CA 1.0	0.20	2.85	−0.73	−4.34	−4.52	0.67	3.12	10.08	0.03
CA 2.5	1.32	4.28	0.10	−5.27	−6.38	−0.21	4.15	11.27	1.32
CA 5.0	2.13	5.10	0.84	−6.33	−7.52	−1.84	5.00	11.74	2.24
CA 7.5	2.84	5.61	2.41	−8.59	−9.24	−3.61	5.52	12.47	4.36
CA 10	3.96	6.12	3.66	−9.38	−11.10	−6.37	6.63	13.03	5.10

**Table 5 foods-13-01455-t005:** MIC values (mg/mL).

	MASP	CP	PT	MASRJ	CLRJ	CLJ	MASA	CLA	AP
*S. pyogenes ATCC 19615*	-	-	-	2.5	5	5	-	5	-
*S. aureus ATCC 25923*	1	-	-	0.25	1	0.25	0.25	-	0.25
*S. flexneri ATCC 12022*	0.25	-	0.25	0.25	0.25	0.25	0.25	-	0.25
*P. aeruginosa ATCC 27853*	-	-	-	0.25	0.75	5	0.25	-	0.25
*E. coli ATCC 25922*	1	-	0.25	0.25	0.25	0.25	0.25	-	0.25
*S. typhimurium ATCC 14028*	0.25	-	0.25	0.25	0.25	0.25	0.25	-	0.25
*H. influenzae ATCC 10211*	0.25	-	-	0.25	0.25	0.25	0.25	-	0.25
*B. cereus ATCC 10876*	-	-	-	0.25	0.5	5	0.25	-	0.25
*Cl perfringens ATCC 13124*	-	-	-	0.25	1	5	0.25	-	0.25
*L. monocytogenes ATCC 19114*	0.5	-	7.5	0.25	0.25	0.25	0.25	-	0.25
*C. parapsilopsis ATCC 22019*	2.5	-	-	0.25	0.25	0.25	0.25	-	0.25
*C. albicans ATCC 10231*	-	-	-	7.5	2.5	7.5	-	10	-
*S. aureus (MLIRP 052022)*	-	-	-	7.5	5	10	7.5	2.5	5
*S. aureus (MLIRP 072022)*	-	-	-	5	5	2.5	2.5	0.25	5
*S. aureus (MLIRP 092020)*	10	-	-	-	-	-	1	0.25	1
*P. aeruginosa (MLIRP 122021)*	0.25	0.25	0.25	-	0.25	-	-	0.25	-
*P. aeruginosa (MLIRP 092022)*	-	-	-	-	0.25	0.25	0.25	0.25	-
*P. aeruginosa (MLIRP 042019)*	0.25	0.25	0.25	0.25	0.25	0.25	0.25	0.25	0.25
*E. coli (MLIRP 062019)*	-	-	-	-	-	0.25	5	0.75	1
*E. coli (MLIRP 022020)*	-	-	-	0.25	0.25	0.25	0.25	0.25	0.25
*E. coli (MLIRP 112020)*	0.25	0.25	0.25	0.25	0.25	0.25	0.5	0.25	0.5

**Table 6 foods-13-01455-t006:** The percentage of inhibition of haemolysis and inhibition of protein denaturation.

Samples	The Concentrations of the Samples							
	2.5%	5%	7.5%	10%	2.5%	5%	7.5%	10%
	% Inhibition of Haemolysis Values				% Inhibition of Protein Denaturation			
MASP	47.22	52.78	57.28	58.83	19.81	82.77	83.23	83.32
CP	0.75	3.41	44.02	51.46	28.40	34.30	42.29	50.61
PT	-	-	0.84	18.71	10.98	8.64	0.77	-
MASRJ	-	-	-	-	34.43	18.75	8.22	7.40
CLRJ	41.56	47.2	55	59.58	81.17	71.18	65.82	65.03
CLJ	-	-	-	14.37	58.42	53.48	51.48	51.42
MASA	-	-	-	-	49.03	38.13	34.69	9.81
CLA	-	-	-	-	45.63	30.35	14.07	13.99
AP	-	-	-	44.72	58.83	58.05	51.15	50.33
CTR-dexamethasone 0.1 mg/mL	61.28				67.18%			

**Table 7 foods-13-01455-t007:** Binding energy interaction between the identified compounds and bacterial protein targets.

S/no	Compounds	Pubchem CID	1JII		1G27		3LPX	
			Binding Energy (kcal/mol)	Number of Bond Interactions	Binding Energy (kcal/mol)	Number of Bond Interactions	Binding Energy (kcal/mol)	Number of Bond Interactions
1	β-resorcylic acid	1491	−6.7	H: 1, π-H: 1 Unfavourable Acceptor: 1	−5.8	H: 3, π-π Stacked: 1, π-Sigma/Alkyl: 1/1	−5.9	H: 3, π-π T-Shaped: 1, Unfavourable Donor: 1
2	Caffeic acid	689043	−7.0	H: 5, π-Alkyl: 1	−6.3	H: 3, π-Sigma: 2, Unfavourable Acceptor: 1	−6.2	H: 5, π-Sulfur/Alkyl: 1/1
3	Epicatechin	72276	−8.3	H: 2, C-H: 1, Unfavourable Donor/π anion: 1	−7.8	H: 4, π-π Stacked: 1, π-Alkyl: 1, Unfavourable Acceptor: 1	−7.5	H: 4, π-Sulfur/Alkyl: 1/1
4	Ferulic acid	445858	−6.5	C-H: 2, Alkyl: 1, Amide-π: 1	−6.2	H: 5, π-Sigma: 1	−6.0	H: 3, C-H: 1, π-Sulfur/Alkyl: 1/1
5	Gallic acid	370	−7.2	H: 7, Amide π-Stacked: 1	−5.9	H: 3, π-H/Sigma: 1/1, Unfavourable Acceptor: 1	−6.1	H: 2, π-π T-Shaped: 1, π-Alkyl: 1
6	p-coumaric	637542	−6.2	H:2, C-H: 1, π-Alkyl:1	−5.9	H: 3, C-H: 1, π-Sigma: 1	−6.0	H: 2
7	Quercetin	5280343	−9.6	H: 3, π-Anion: 1, Unfavourable Donor: 2	−7.5	H: 1, C-H: 1, Π-Anion/Alkyl/Sigma: 1/2/1, π-π Stacked: 1, Unfavourable Donor: 1	−7.9	H: 4, C-H: 2, π-Sulfur/Sigma/Alkyl: 1/2/1
8	Resveratrol	445154	−8.2	H: 5, π-anion/alkyl: 2, Unfavourable Donor: 1	−6.5	H: 1, π-Anion:1, π-π-Stacked: 1, π-Sigma/Alkyl: 1/1	−7.4	H: 2, π-π T Shaped: 1, π-Sulfur/Alkyl: 1/1, Unfavourable Donor: 1
9	Rosmarinic acid	5281792	−8.0	H: 4, π-Anion: 1	−8.2	H: 7, π-π Stacked: 1, Π-Sigma/Alkyl: 1/1	−7.6	H: 6, C-H: 1, Π-Π T-Shaped: 1, Π-Alkyl: 3
10	Rutin	5280805	−9.7	H: 6, C-H: 1, π-H: 3, π-Anion/Alkyl: 1/1, Unfavourable Donor: 1	−8.3	van der Waals: 1, H: 6, Amide-π: 1, π -Alkyl: 2, Unfavourable Donor: 1	−9.0	H: 6, π-π T Shaped: 1, π-Alkyl: 2, Unfavourable Donor: 1

1JII: Tyrosyl-tRNA synthetase; IG27: Peptide deformylase; 3LPX: DNA gyrase; H: hydrogen bond; C-H: carbon-hydrogen bond; π: pi.

**Table 8 foods-13-01455-t008:** Binding energy interaction between the identified compounds and fungal protein targets.

S/no	Compounds	Pubchem CID	2QA1		8JZN		5AG7	
			Binding Energy (kcal/mol)	Number of Bond Interactions	Binding Energy (kcal/mol)	Number of Bond Interactions	Binding Energy (kcal/mol)	Number of Bond Interactions
1	β-resorcylic acid	1491	−6.1	H: 4, π-Alkyl: 2, Unfavourable Donor: 1	−6.6	H: 2, π-π Stacked: 1	−6.3	H: 1, π-π T Shaped: 1, π-Sigma/Alkyl: 1/1
2	Caffeic acid	689043	−6.1	H: 3, π-Alkyl/Sigma: 1/1	−6.5	H: 2, π-Sigma/Anion/Alkyl: 1/1/1	−7.5	H: 1, π-Sigma/Alkyl: 1/3, π-π T Shaped: 2
3	Epicatechin	72276	−8.0	H: 5, Amide-π Stacked: 1, π-π T-Shaped: 1, π-Alkyl: 1	−8.7	H: 3, π-Anion/Cation/Sulfur/Alkyl: 1/1/1/2	−8.1	H: 2, C-H: 1, π-Alkyl: 3
4	Ferulic acid	445858	−5.9	H: 3, C-H: 1, Alkyl: 1	−6.8	H: 3, π-Anion/Sigma: 1/1, Alkyl/π-Alkyl: 1/1	−7.0	H: 1, π-Sigma: 1, π-π T Shaped: 1, Alkyl/π-Alkyl: 4
5	Gallic acid	370	−6.8	H: 5, C-H: 1, π-Alkyl: 2	−6.9	H: 2, π-Sigma/Alkyl/Anion: 1/1/1, Unfavourable Donor: 1	−6.5	H: 5, π-Sigma/Alkyl: 1/1
6	p-coumaric	637542	−6.1	H: 2, π-Alkyl: 3	−6.6	H: 3, π-Sigma/Anion/Alkyl: 1/1/1	−7.6	H: 1, π-Sigma/Alkyl: 2/2, π-π T Shaped: 1
7	Quercetin	5280343	−7.9	H: 4, Unfavourable Acceptor: 1	−8.7	H: 1, π-π T Shaped: 1, π-Sigma/Alkyl: 2/5	−9.2	H: 2, C-H: 1, π-Anion/Sigma/Alkyl: 1/1/3, Unfavourable Donor/Acceptor: 1/1
8	Resveratrol	445154	−7.8	H: 1, π-Alkyl: 2	−7.8	H: 1, π-Alkyl: 4, π-π T Shaped: 1	−9.5	H: 1, π-Sigma: 4, π-π/Stacked/T Shaped: 1/1, π-Alkyl: 3
9	Rosmarinic acid	5281792	−8.1	H: 4, π-Sigma/Alkyl: 1/1, Unfavourable Donor: 1	−8.6	H: 3, π-Sigma/Alkyl: 1/5	−8.3	H: 3, π-π T Shaped: 2, Unfavourable Donor: 1
10	Rutin	5280805	−9.4	H: 1, π-Sigma/Alkyl: 1/1, π-π Stacked: 1, Unfavourable Donor/Acceptor: 1/1	−9.4	H: 5, C-H: 2, π-Anion/Alkyl: 2/2	−9.5	H: 3, C-H: 1, π-Sigma/Alkyl: 1/4, π-π Stacked: 1, Unfavourable Acceptor: 1

2QA1: Squalene epoxidase; 8JZN: 1,3-β-glucan synthase; 5AG7: N-myristoyl transferase; H: hydrogen bond; C-H: carbon-hydrogen bond; π: pi.

## Data Availability

The report of the analyses performed for the samples in the paper can be found at the Interdisciplinary Research Platform (PCI) belonging to the University of Life Sciences “King Mihai I” Timisoara.

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
