# Peer review of "Romanian Bee Product Analysis: Chemical Composition, Antimicrobial Activity, and Molecular Docking Insights"

_foods, 2024, doi:10.3390/foods13101455_

Round 1

Reviewer 1 Report

Comments and Suggestions for Authors

The research work has interesting results. However, the analysis of the results is not meaningful. You are not giving a detailed and argued explanation of the findings obtained. In addition, you do not report similar work to compare the results you have obtained. 

You should improve the analysis and interpretation of the results, I attach a word with comments and suggestions. 

Comments on the Quality of English Language

The wording should be revised, some words should be repeated, and some long sentences that make the idea get lost. 

Author Response

Date: Timisoara /30.04.2024

Name: Diana Obistioiu

University:        UNIVERSITY OF LIFE SCIENCES "KING MIHAI I" FROM TIMISOARA,

Address:             Calea Aradului No.119, 300645, Timisoara, Romania

E-mail: dianaobistioiu@usvt.ro

Rebuttal letter

Dear Editor

We would like to express our gratitude for the constructive observations, corrections, and recommendations.

Based on the reviewers’ recommendations, the authors of this paper responded point by point to the following aspects:

  1. Reviewer 1 comments:

The research work has interesting results. However, the analysis of the results is not meaningful. You are not giving a detailed and argued explanation of the findings obtained. In addition, you do not report similar work to compare the results you have obtained.

You should improve the analysis and interpretation of the results, I attach a word with comments and suggestions.

Introduction

Comment: A paragraph on the characteristics of Romanian bees and generalities of production should be included.

Answer: Thank you for the relevant feedback. The paragraph was inserted in the Introduction section.

Comment:  Some examples of similar studies should be included. To know the most relevant results on the composition of bee products.

Answer: This part was also inserted in the modified section of the Introduction and as well as in the revised version of the Discussions.

Materials and methods

Comment: In line 146 the LC-MS acronym must be defined.

Answer: The definition was inserted in the text.

Comment: In section 2.3.1. (Line 181), why were those concentrations chosen?

Answer: The concentrations tested were selected based on previous research and a literature search to cover a broad range of concentrations and find possible MIC values. The explanations were inserted in section 2.3.1.

Results

Comment: The analysis of the results should be completed in section 3.1. It should provide an analysis and explanation of the phenolic compounds and their amounts according to the physicochemical characteristics of each type of bee product. In addition, it should include a comparative analysis between the samples analyzed. Finally, it should include some similar reports to compare the main phenolic compounds and the characteristics of the bee products.

Answer: Thank you very much for your observation. All explanations and comparative literature studies were added in section 4.1 of Discussions.

Comment: The analysis of the results should be completed in section 3.2 (antimicrobial activity). The analysis of antimicrobial activity should be associated with the results obtained in Table 1 (phenolic compounds). It is well known that high activity is associated with high concentrations. In addition, the mechanism of action of the phenolic compounds identified in Gram-negative and Gram-positive bacteria should be described, taking into account that these bacteria have different structures.

Answer: All the suggestions were considered and inserted in section 4.2

Comment: In section 3.3. (Inhibition of hemolysis and protein denaturation values Inhibition of hemolysis values and Inhibition of protein denaturation) the discussion should be completed. Explain the possible mechanism of action of bee products and their protective effect against hemolysis and inhibition of protein denaturation. Support this explanation with the main phenolic compounds identified. Also, mention similar reports demonstrating the same or similar effects when using similar compounds.

Answer: The explanations and similar reports were inserted in section 4.3

Comment: In section 3.4 (Analysis of docking) the analysis of the results should be completed. In the analysis of results, it is necessary to mention and argue the most important aspects such as:

Binding affinity: Describe the strength and nature of the interaction between the molecules, typically expressed as binding energy. Higher binding affinity indicates a stronger interaction and potentially greater biological activity.

Interaction energy: What is the interaction energy between the molecules and how it influences the stability of the formed complex?

Complex conformation: How does the interaction occur and if there are specific interactions, such as hydrogen bonds or hydrophobic interactions?

Complex stability: How stable the formed complex can be?

Answer: Thank you very much for these interesting observations. We have tried to explain the strength of the interactions between our compounds and protein targets, which overhauls that of the initial submission, and the explanations were inserted in the manuscript.

Comment: Moreover, it is important to compare the results obtained with similar reports and other known reference compounds or control compounds to validate the findings and provide a context for interpreting the potential biological activity of the compounds here studied.

Answer: Again, we are grateful for your kind comment and for seeing that our findings corroborate previous research. Hence, we carefully discussed our docking outcomes while relating them to previous authors’ reports with similar findings.

Comment: The wording should be revised, some words should be repeated, and some long sentences that make the idea get lost.

Answer: The manuscript was revised from the language/grammar point of view.

We would like to thank the reviewer for the corrections and recommendations, which contributed to the significant improvement of the paper.

Reviewer 2 Report

Comments and Suggestions for Authors

I found the article about Romanian bee products interesting because it described data not only on propolis and royal jelly but also on drone brood. I didn't find much literature describing drone brood. However, I pointed out below a few details that could/should be corrected to improve the quality of the article itself. 

1/ Standards of polyphenols that were analysed by LC-MS should be listed in the section M&M.

2/ l. 142. I don't think that this information is important when authors already declared their affiliations.

3/ The abbreviations for samples don't sound logical to me e.g. "commercial lyophilised royal jelly LML", my suggestion is lyophilised royal jelly- commercial  LRJ-COM, then "commercial royal jelly LMC" for me: royal jelly commercial RJ-COM, and then "royal jelly from micro apiary source LMS", my point is: royal jelly from micro-apiary RJ-MA.

missing lyophilised royal jelly from the micro-apiary?

4/ Table 1, please work on the table look, and put also meaning of abbreviations below each table. Should be written that the differences were detected in the column. Moreover, add below the statistical test that was used.

5/Add explanations of abbreviations below figures. Also, if possible decrease the number of figures in the main text, the most interesting keep in the main text, the rest try to move to supplementary materials. The same situation appeared for molecular docking. The most relevant are kept in the main text and the others are kept in the supplementary.

6/ The "Discussion" is not written in a good manner, it is mostly reporting results from literature. It doesn't discuss the obtained results with literature and why the difference appeared in the chapter on chemical composition. it is a little bit better in discussion of the inhibition of hemolysis. 

7/How the results refer to molecular docking, I didn't find a discussion for molecular docking itself. 

Comments on the Quality of English Language

required extensive English editing.

Author Response

Date: Timisoara /30.04.2024

Name: Diana Obistioiu

University:        UNIVERSITY OF LIFE SCIENCES "KING MIHAI I" FROM TIMISOARA,

Address:             Calea Aradului No.119, 300645, Timisoara, Romania

E-mail: dianaobistioiu@usvt.ro

Rebuttal letter

Dear Editor

We are grateful for the constructive observations, corrections, and recommendations.

Based on the reviewers’ recommendations, the authors of this paper responded point by point to the following aspects:

  1. Reviewer 2 comments:

I found the article about Romanian bee products interesting because it described data not only on propolis and royal jelly but also on drone brood. I didn’t find much literature describing drone brood. However, I pointed out below a few details that could/should be corrected to improve the quality of the article itself.

Comment: 1/ Standards of polyphenols that were analysed by LC-MS should be listed in the section M&M.

Answer: The standards were added in the Material and Methods section 2.2.

Comment: 2/ l. 142. I don’t think that this information is important when authors already declared their affiliations.

Answer: Thank you for this observation; we have made the necessary modifications.

Comment: 3/ The abbreviations for samples don’t sound logical to me e.g. “commercial lyophilised royal jelly LML”, my suggestion is lyophilised royal jelly- commercial  LRJ-COM, then “commercial royal jelly LMC” for me: royal jelly commercial RJ-COM, and then “royal jelly from micro apiary source LMS”, my point is: royal jelly from micro-apiary RJ-MA.

missing lyophilised royal jelly from the micro-apiary?

Answer: We have modified the text per your suggestion. Regarding the lyophilised royal jelly from the micro-apiary, samples were selected to include only those available on the market.

Comment: 4/ Table 1, please work on the table look, and put also meaning of abbreviations below each table. Should be written that the differences were detected in the column. Moreover, add below the statistical test that was used.

Answer: Thank you for your observations; the Table was modified as well and the abbreviations were inserted as the table footer.

Comment: 5/Add explanations of abbreviations below figures. Also, if possible decrease the number of figures in the main text, the most interesting keep in the main text, the rest try to move to supplementary materials. The same situation appeared for molecular docking. The most relevant are kept in the main text and the others are kept in the supplementary.

Answer: We sincerely appreciate your kind comment; the meaning of the abbreviations in Table 3 and Table 4 have been provided. As for the docking figures, they were selected based on the best interaction the compounds possess with the protein targets, and we thought it could be better shown to the reader in the main text. The figures from the microbiology analysis were removed and inserted as supplementary files. All the data was added in the manuscript as a table.

Comment: 6/ The “Discussion” is not written in a good manner, it is mostly reporting results from literature. It doesn’t discuss the obtained results with literature and why the difference appeared in the chapter on chemical composition. it is a little bit better in discussion of the inhibition of hemolysis.

Answer: Thank you very much for the observation. The discussion section was rewritten.

Comment: 7/How the results refer to molecular docking, I didn’t find a discussion for molecular docking itself.

Answer: Thank you for this observation, we have provided a convincing discussion of our molecular docking outcome.

Comment: Required extensive English editing.

Answer: The English writing was revised.

We want to thank the reviewer for the corrections and recommendations, which contributed to the significant improvement of the paper.

Round 2

Reviewer 2 Report

Comments and Suggestions for Authors

English has been improved.